# Theoretical and Numerical Investigation of Damage Sensitivity of Steel–Concrete Composite Beam Bridges

Zhibo Guo [1], Jianqing Bu [2,3,*], Jiren Zhang [4,*], Wenlong Cao [5] and Xiaoming Huang [6]

1 School of Civil Engineering, Shijiazhuang Tiedao University, Shijiazhuang 050043, China; guozhibo19991010@163.com
2 State Key Laboratory of Mechanical Behavior and System Safety of Traffic Engineering Structures, Shijiazhuang Tiedao University, Shijiazhuang 050043, China
3 School of Traffic and Transportation, Shijiazhuang Tiedao University, Shijiazhuang 050043, China
4 Key Laboratory of Wind and Bridge Engineering of Hunan Province, College of Civil Engineering, Hunan University, Changsha 410082, China
5 Department of Road and Bridge Engineering, Hebei Jiaotong Vocational and Technical College, Shijiazhuang 050091, China; caowenlong@hejtxy.edu.cn
6 School of Transportation, Southeast University, Nanjing 210096, China; huangxm@seu.edu.cn
* Correspondence: bujq@stdu.edu.cn (J.B.); jrzhang1994@hnu.edu.cn (J.Z.)

**Abstract:** To investigate the sensitivity of the overall mechanical performance of steel–concrete composite beam bridges (SCCBBs) to different types of damage, this paper proposes a method of analyzing the sensitivity of SCCBBs to damage based on the extremely randomized trees (ET) algorithm in machine learning. A steel–concrete composite continuous beam bridge was used as the engineering basis, and the finite element method was used to analyze the changes in the static and dynamic response of the bridge caused by seven types of damage. The proposed SCCBB damage sensitivity analysis theory was used to explore the sensitivity factors of the seven types of damage. The results show that microcracks in steel beams have the most significant impact on the mechanical performance sensitivity of SCCBBs, followed by the concrete slab stiffness degradation and bridge deck breakage. The sensitivity of the damage caused by transverse diaphragms and bridge pier stiffness degradation is relatively low, while the sensitivity of stud fractures and bearing damage is minimal. The impact factors of damage sensitivity were 0.51, 0.19, 0.13, 0.08, 0.05, 0.03 and 0.01. This research can provide a reference for the damage classification of SCCBBs with multiple damage interlacing.

**Keywords:** steel–concrete composite beam bridge; structural damage; sensitivity analysis; vehicle–bridge interaction

## 1. Introduction

With the rapid development of transportation infrastructure construction in China, SCCBBs have been widely used in the construction of highways and urban elevated bridges due to their advantages, such as fast construction speeds and an attractive appearance. These structures offer significant technical and socio-economic benefits [1,2]. However, over the long term, the safety and durability of SCCBBs are subject to severe challenges due to the natural aging of materials, cyclic effects of complex loads, and external unfavorable factors such as natural disasters and human-induced incidents [3,4].

Currently, scholars both domestically and internationally have conducted extensive research on the mechanical behavior and damage mechanism of SCCBBs. Research on the dynamic response of SCCBBs has been relatively in-depth, particularly considering the impact of vehicle dynamic loads on these structures. However, constructing dynamic models of steel–concrete composite beams (SCCBs) and dynamic response calculation methods that consider interfacial slip remain a hot research topic. Lin et al. [5,6] developed

a dynamic finite element framework that considers interface slip damage between concrete slabs and steel beams. This framework is based on the dynamic variational principle and Hamilton's principle. They investigated the dynamic responses of composite beam systems under various complex loading conditions. Li et al. [7] analyzed the free vibration of SCCBs using the dynamic stiffness method and derived the coupled vibration equation of SCCBs through Hamilton's principle. These studies have further advanced the theoretical application of dynamic response algorithms for SCCBs.

Throughout the service lives of SCCBBs, various adverse factors, such as environmental erosion and vehicle loads, inevitably result in varying degrees of damage to the bridge components. As a load-bearing component that connects concrete slabs and steel beams, the post-damage shear resistance, fatigue performance, and frictional performance of studs have garnered widespread attention from both domestic and foreign scholars [8–14]. Furthermore, investigating the fatigue damage mechanism of bridge deck slabs and steel beams is also a research hotspot [15]. Orthotropic steel bridge deck panels are lightweight and have excellent aerodynamic performance, but they are prone to fatigue damage. Previous research has demonstrated that utilizing Engineered Cementitious Composites (ECC) and large U-ribs can effectively improve the fatigue performance of orthotropic steel bridge deck panels [16], and Gao et al. [17] conducted an analysis of the fatigue performance of combination box girders with varying reinforcement ratios under complete shear connection. Additionally, both domestic and foreign scholars aim to optimize fatigue damage evolution simulation algorithms and improve computational efficiency for large-span steel bridges [18].

The frequent occurrence of natural and man-made disasters is also a significant factor that contributes to the damage of SCCBBs [19–21]. Zhu et al. [22,23] performed a structural analysis of steel–concrete composite bridge beams (SCCBBs) under various explosion loads and source forms using the completely multi-Euler domain method and Autodyn commercial software to investigate their mechanical properties. Yun et al. [24] evaluated the fire damage and structural performance of the upper structure of highway SCCBBs using a fluid–structure coupling fire analysis method. Wang et al. [25] measured and evaluated the impact pressure and dynamic characteristics of mudflow on scaled bridge piers. Amadio et al. [26] proposed an improved finite element method (FEM) to predict the overall and local behavior of steel–concrete composite joints under seismic loads. Additionally, the vulnerability of bridge structures to the impact of floods is also a current research hotspot [27]. In the field of bridge damage detection technology, Tung et al. [28] developed a completely non-contact health detection system framework based on conventional camera and computer vision technology to obtain the structural vibration displacement. Acevedo et al. [29] used accelerometers and smartphones to determine the acceleration in three directions of bridges. In addition, they used only sensors to perform fast Fourier transform (FFT) on the structural responses, and the results verified the applicability of fusion-only sensors in the health detection of real bridge structures. In the face of the huge stock of sub-healthy bridges and the development of bridge damage detection technology, the use of different equipment and technology to detect bridge structural damage is also the focus of scholars at home and abroad [28–32].

Sensitivity evaluation has been a crucial step in disaster prevention and reduction, attracting the attention of researchers from various countries for many years [33]. In the field of machine learning research, Wang [34] used an ET algorithm to construct a mapping relationship between seismic parameters and structural damage indicators, providing a theoretical reference for regional seismic damage assessment. To evaluate the potential destructive power of seismic motion, Wu et al. [35] revealed the key seismic intensity indicators affecting the destructive power through machine-learning-based sensitivity analysis methods. Shao et al. [36,37] constructed landslide sensitivity evaluation models using various machine learning algorithms. These studies further promote the integration of machine learning and bridge engineering.

As a matter of fact, the SCCBBs suffering from natural disasters, man-made disasters, or long-term service are interwoven with multiple damage types, and the structural state

of the bridge is complex. Different damage types have different effects on the overall mechanical properties of SCCBBs, and different damage degrees of a single damage type have different effects on the overall mechanical properties of SCCBBs. For this reason, there is no clear quantitative evaluation standard for the damage grading of SCCBBs. At present, domestic and foreign scholars rarely study the damage sensitivity of composite structure bridges. Therefore, quantifying the sensitivity of different damages to the overall mechanical properties of SCCBBs and then determining the weight influence values of different damage types is of great significance for the establishment of damage grade classification standards for SCCBBs. It can provide a reference for the design, construction, and testing of SCCBBs.

Based on existing research results, this paper derives the dynamic equations of the three-axle vehicle–bridge coupling system using the d'Alembert principle. A damage sensitivity analysis method for SCCBBs is proposed based on the ET algorithm. Taking a three-span steel–concrete composite continuous beam bridge as an engineering example, dynamic indicators such as the frequency of the bridge structure, displacement, and acceleration responses at the side span and midspan during vehicle passing are selected, as well as static indicators such as the maximum deflection, stress, and pier top displacement under dual-lane loading. The sensitivity influencing factors of seven types of damage, such as bridge deck damage, concrete slab stiffness degradation, and stud fracture, are analyzed to provide a reference for the research of disaster damage classification standards for SCCBBs.

## 2. Vehicle–Bridge Coupling Vibration Model

### 2.1. Vehicle Dynamics Equation

In the field of research on vehicle–bridge coupling vibration, commonly used vehicle models include the single-wheel vehicle model, half-vehicle model, and spatial full-vehicle model [38,39]. In order to accurately characterize the vehicle model, this paper adopts the three-axle spatial full-vehicle model from the American Association of State Highway and Transportation Officials (AASHTO) bridge design specification HS20-44 to establish the motion differential equations of the vehicle. The vehicle has a total of 12 degrees of freedom, which include the vertical displacement of each of the 6 wheels, as well as the vertical displacement, roll angle, and pitch angle of the front and rear of the vehicle. In the vehicle model, each tire and its connecting parts to the chassis are simulated using two spring–damper systems and a mass point model. The masses of the front and rear of the vehicle are concentrated at the center of gravity, and both the front and rear are considered rigid bodies, as shown in Figure 1.

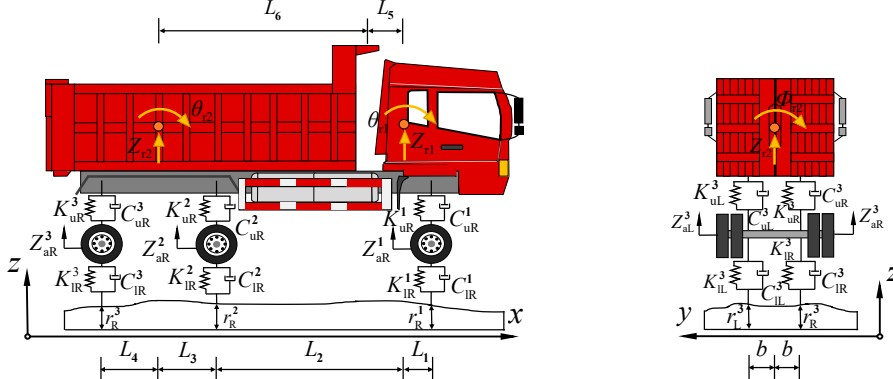

**Figure 1.** Mass–spring–damper model of AASHTO HS20-44 vehicle.

The three-axle vehicle is discretized into several mass–spring–damper models, and the freedom vector of the vehicle is

$$d_{\mathrm{v}} = \left[ Z_{\mathrm{aL}}^1,\ Z_{\mathrm{aR}}^1,\ Z_{\mathrm{aL}}^2,\ Z_{\mathrm{aR}}^2,\ Z_{\mathrm{aL}}^3,\ Z_{\mathrm{aR}}^3,\ Z_{\mathrm{r1}},\ \Phi_{\mathrm{r1}},\ \theta_{\mathrm{r1}},\ Z_{\mathrm{r2}},\ \Phi_{\mathrm{r2}},\ \theta_{\mathrm{r2}} \right] \tag{1}$$

where $Z_{aL}^i$ and $Z_{aR}^i$ ($i$ = 1, 2, 3) are the vertical displacements of the tires on the left and right sides. $Z_{r1}$ and $Z_{r2}$ are the vertical displacements of the front and rear of the vehicle. $\theta_{r1}$ and $\theta_{r2}$ are the pitch angle displacements of the front and rear of the vehicle, respectively. $\Phi_{r1}$ and $\Phi_{r2}$ are the lateral tilt angle displacements of the front and rear of the vehicle, respectively [40].

The mass (moment of inertia) vector of the vehicle is

$$M = \left[ M_{aL}^1,\ M_{aR}^1,\ M_{aL}^2,\ M_{aR}^2,\ M_{aL}^3,\ M_{aR}^3,\ M_{r1},\ J_{yz}^1,\ J_{zx}^1,\ M_{r2},\ J_{yz}^2,\ J_{zx}^2 \right] \tag{2}$$

where $M_{aL}^i$ and $M_{aR}^i$ ($i$ = 1, 2, 3) are the tire masses of the left and right sides of the vehicle, and $M_{r1}$ and $M_{r2}$ are the front and rear of the vehicle masses, respectively. $J_{yz}^i$ ($i$ = 1, 2) is the moment of inertia of the vehicle body roll characteristics. $J_{zx}^i$ ($i$ = 1, 2) is the moment of inertia of the pitching characteristics of the vehicle body.

The vehicle stiffness coefficient vector is

$$K = \left[ K_{uL}^1,\ K_{uR}^1,\ K_{uL}^2,\ K_{uR}^2,\ K_{uL}^3,\ K_{uR}^3,\ K_{lL}^1,\ K_{lR}^1,\ K_{lL}^2,\ K_{lR}^2,\ K_{lL}^3,\ K_{lR}^3 \right] \tag{3}$$

where $K_{uL}^i$ and $K_{uR}^i$ ($i$ = 1, 2, 3) are the suspension stiffness coefficients of the left and right sides of the vehicle. $K_{lL}^i$ and $K_{lR}^i$ ($i$ = 1, 2, 3) are the tire stiffness coefficients of the left and right sides of the vehicle.

The damping coefficient vector of the vehicle is

$$C = \left[ C_{uL}^1,\ C_{uR}^1,\ C_{uL}^2,\ C_{uR}^2,\ C_{uL}^3,\ C_{uR}^3,\ C_{lL}^1,\ C_{lR}^1,\ C_{lL}^2,\ C_{lR}^2,\ C_{lL}^3,\ C_{lR}^3 \right] \tag{4}$$

where $C_{uL}^i$ and $C_{uR}^i$ ($i$ = 1, 2, 3) are the suspension damping coefficients of the left and right sides of the vehicle. $C_{lL}^i$ and $C_{lR}^i$ ($i$ = 1, 2, 3) are the tire damping coefficients of the left and right sides of the vehicle.

Assume that the displacement at the contact position between the wheel and the bridge (including the deck irregularity) is

$$Y = \left[ y_L^1, y_R^1, y_L^2, y_R^2, y_L^3, y_R^3 \right] \tag{5}$$

where $y_L^i$ ($i$ = 1, 2, 3) denotes the displacements of the left-side three wheel contact points of the vehicle on the bridge, while $y_R^i$ ($i$ = 1, 2, 3) corresponds to those of the right-side three contact points.

This paper adopts the direct equilibrium method (d'Alembert's principle) to establish the equations of motion for the three-dimensional (3D) vehicle model in the AASHTO Bridge Design Specification HS20-44, as shown in Equation (6) [41].

$$M_v \ddot{d}_v + C_v \dot{d}_v + K_v d_v = F_{vb} + f_{vg} \tag{6}$$

where $M_v$, $C_v$, and $K_v$ are the mass, damping, and stiffness matrices of the vehicle structure, respectively, all of which are 11th-order matrices, as shown in Equations (7)–(9). The submatrices in $C_v$ and $K_v$ are shown in Appendix B. $d_v$, $\dot{d}_v$, and $\ddot{d}_v$ are the displacement, velocity, and acceleration arrays of the vehicle, respectively. $F_{vb}$ is the load vector of the vehicle–bridge interaction force caused by wheel deformation, and $f_{vg}$ is the load vector caused by the self-weight of the vehicle, both of which are 11th-order vectors, as shown in Equation (10).

$$M_v = \begin{bmatrix} M_{aL}^1 & & & & & & & & \\ & M_{aR}^1 & & & & & & & \\ & & \ddots & & & & & & \\ & & & M_{aR}^3 & & & & & \\ & & & & M_{r1} & 0 & 0 & M_{r2} & 0 \\ & & & & 0 & J_{yz}^1 & 0 & 0 & 0 \\ & & & & -L_5 M_{r1} & 0 & J_{zx}^1 & 0 & 0 \\ & & & & -\frac{M_{r1}L_6 + J_{zx}^2}{L_6} & 0 & -\frac{L_5 J_{zx}^2}{L_6} & \frac{J_{zx}^2}{L_6} & 0 \\ & & & & 0 & 0 & 0 & 0 & J_{yz}^2 \end{bmatrix} \tag{7}$$

$$K_v = \begin{bmatrix} K_{11_{6\times6}} & K_{12_{6\times5}} \\ K_{21_{5\times6}} & K_{22_{5\times5}} \end{bmatrix} \tag{8}$$

$$C_v = \begin{bmatrix} C_{11_{6\times6}} & C_{12_{6\times5}} \\ C_{21_{5\times6}} & C_{22_{5\times5}} \end{bmatrix} \tag{9}$$

$$F_{vb} = \begin{bmatrix} K_{1L}^1 y_L^1 + C_{1L}^1 \dot{y}_L^1 \\ K_{1R}^1 y_R^1 + C_{1R}^1 \dot{y}_R^1 \\ K_{1L}^2 y_L^2 + C_{1L}^2 \dot{y}_L^2 \\ K_{1R}^2 y_R^2 + C_{1R}^2 \dot{y}_R^2 \\ K_{1L}^3 y_L^3 + C_{1L}^3 \dot{y}_L^3 \\ K_{1R}^3 y_R^3 + C_{1R}^3 \dot{y}_R^3 \\ 0 \\ 0 \\ 0 \\ 0 \\ 0 \end{bmatrix}, \quad f_{vg} = - \begin{bmatrix} M_{aL}^1 \\ M_{aR}^1 \\ M_{aL}^2 \\ M_{aR}^2 \\ M_{aL}^3 \\ M_{aR}^3 \\ M_{r1} + M_{r2} \\ 0 \\ -L_5 M_{r1} \\ -L_6 M_{r1} \\ 0 \end{bmatrix} g \tag{10}$$

### 2.2. Dynamic Equation of Vehicle–Bridge Coupling Vibration

When establishing the dynamic equations of the entire drivetrain coupling system, the motion equations of the three-axle vehicle can be expressed as in Equation (6), while the motion equations of the bridge can be expressed as

$$M_b \ddot{d}_b + C_b \dot{d}_b + K_b d_b = F_{bv} \tag{11}$$

where $M_b$, $C_b$, and $K_b$ are, respectively, the mass matrix, damping matrix, and stiffness matrix of the bridge. $d_b$ is the displacement matrix of the bridge. $F_{bv}$ is the moment matrix of the forces applied by a three-axis vehicle on the bridge deck.

Based on the displacement coordination and interaction force relationship at the contact point between the three-axis vehicle and the bridge, Equations (6) and (11) can be combined to obtain the vibration equation of the vehicle–bridge coupling system [41,42], as shown in Equation (12).

$$\begin{bmatrix} M_b & \\ & M_v \end{bmatrix} \begin{bmatrix} \ddot{d}_b \\ \ddot{d}_v \end{bmatrix} + \begin{bmatrix} C_b + C_{b-b} & C_{b-v} \\ C_{v-b} & C_v \end{bmatrix} \begin{bmatrix} \dot{d}_b \\ \dot{d}_v \end{bmatrix} + \begin{bmatrix} K_b + K_{b-b} & K_{b-v} \\ K_{v-b} & K_v \end{bmatrix} \begin{bmatrix} d_b \\ d_v \end{bmatrix} = \begin{bmatrix} F_{b-r} \\ F_{v-r} + F_G \end{bmatrix} \tag{12}$$

where $F_G$ is the gravity of the vehicle; $C_{b-b}$, $C_{b-v}$, $C_{v-b}$, $K_{b-b}$, $K_{b-v}$, $K_{v-b}$, $F_{b-r}$, and $F_{v-r}$ are the contact forces between the wheels and the bridge deck that vary over time.

### 3. Vehicle–SCCBB Finite Element Model

In the early studies of vehicle–bridge coupling vibration, the planar beam–rod model or the spatial beam–lattice model was often used. The planar beam–rod model could not

consider the torsion of the bridge, and the spatial beam–lattice model could not obtain the local dynamic response of the bridge. The ANSYS finite element software can be used to establish the 3D solid model of the bridge, and the steel–concrete composite beam bridge is composed of three different components: concrete slab, peg, and steel beam. In order to consider the combination effect of these three components more comprehensively, ANSYS modeling is necessary [43]. A finite element model of the steel–concrete composite beam bridge can be established using solid elements in ANSYS, and a connector of the steel–concrete composite interface can be established using spring elements [44]. The corresponding degrees of freedom of the spring elements can be released to simulate the sliding effect.

### 3.1. Bridge Model

Based on the theoretical framework of vehicle–bridge coupling vibration in Section 2.1, finite element models of three-span steel–concrete composite continuous beam bridges with seven damage types (each containing five damage grades) were established using the ANSYS software. The bridge span was arranged in a 3 × 40 m pattern, the bridge width was 2 × 12.75 m, and the vehicle load rating was highway class I, with a one-way two-lane design speed of 100 km/h. The specific technical indicators and material characteristics are available in the "General Drawing of Prefabricated I-Type Composite Beam Bridge" JTG/T 3911-02-2021 [45]. In the finite element model of the bridge built with the ANSYS software [46], the concrete bridge deck and piers were modeled using the 3D solid element SOLID45, the steel beams using shell element SHELL63, the studs using 3D spring element COMBIN14, and the bearings using a combined simulation of the COMBIN14 and COMBIN40 elements, as illustrated in Figure 2. The number of element types is shown in Table 1. The whole finite element model of the bridge has 90,920 nodes. In the bridge model, a hexahedral mesh is used for the solid element, and a square mesh is used for the shell element. The mesh side length is 0.3 m.

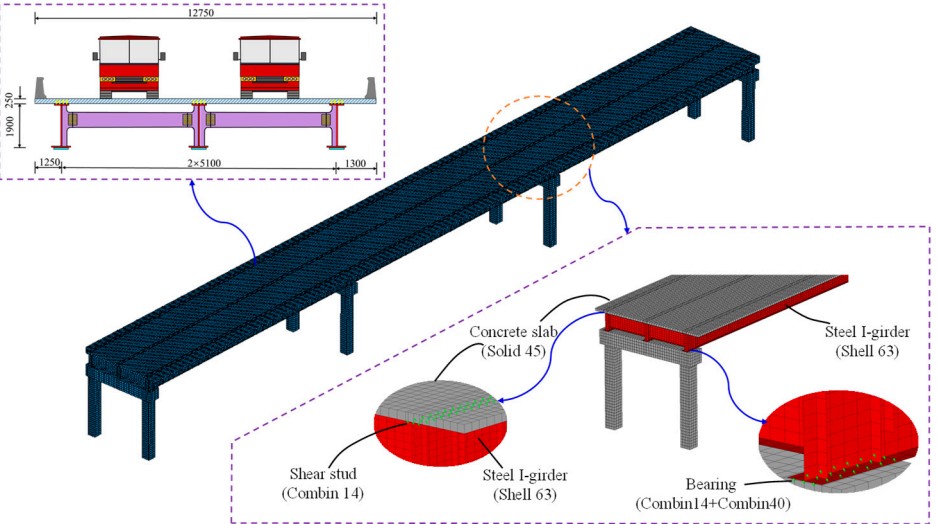

**Figure 2.** Finite element model of the bridge.

**Table 1.** Summary of the number of elements (PCS).

| Element Type | SHELL63 | SOLID45 | COMBIN14 | COMBIN40 | Total |
|:---:|:---:|:---:|:---:|:---:|:---:|
| **Number** | 29,614 | 32,496 | 3954 | 273 | 66,610 |

### 3.2. Vehicle Model

The vehicle finite element model was established using the three-axis spatial whole vehicle model outlined in Section 2.1 of the American Association of State Highway and

Transportation Officials (AASHTO) Bridge Design Specification HS20-44, as depicted in Figure 1. The parameter values for the vehicle model can be found in Table 2 [40]. Within the vehicle finite element model, the vehicle body is represented by the MPC184 rigid rod element, while the vehicle body mass and wheel mass are represented by the MASS21 element. The suspension and vehicle body connection components are represented by the COMBIN14 element, and the wheel and suspension connection components also adopt the COMBIN14 element.

**Table 2.** Vehicle model parameter.

| Parameter | Value |
|---|---|
| Body 1 mass ($M_{r1}$/kg) | 2611.8 |
| Pitch moment of inertia ($J_{zx}^1$/kg·m$^2$) | 2022 |
| Roll moment of inertia ($J_{yz}^1$/kg·m$^2$) | 8544 |
| Body 1 mass ($M_{r2}$/ kg) | 26,113 |
| Pitch moment of inertia ($J_{zx}^1$/kg·m$^2$) | 33,153 |
| Roll moment of inertia ($J_{yz}^1$/kg·m$^2$) | 181,216 |
| Axle 1 mass ($M_{aL}^1$, $M_{aR}^1$/kg) | 245 |
| Suspension stiffness ($K_{uL}^1$, $K_{uR}^1$/kN·m$^{-1}$) | 243 |
| Suspension damping ($C_{uL}^1$, $C_{uR}^1$/kN·s·m$^{-1}$) | 2.19 |
| Tire stiffness ($K_{lL}^1$, $K_{lR}^1$/kN·m$^{-1}$) | 875.08 |
| Tire damping ($C_{lL}^1$, $C_{lR}^1$/kN·s·m$^{-1}$) | 2 |
| Axle 2 mass ($M_{aL}^2$, $M_{aR}^2$/kg) | 405 |
| Suspension stiffness ($K_{uL}^2$, $K_{uR}^2$/kN·m$^{-1}$) | 1903.17 |
| Suspension damping ($C_{uL}^2$, $C_{uR}^2$/kN·s·m$^{-1}$) | 7.88 |
| Tire stiffness ($K_{lL}^2$, $K_{lR}^2$/kN·m$^{-1}$) | 3503.31 |
| Tire damping ($C_{lL}^2$, $C_{lR}^2$/kN·s·m$^{-1}$) | 2 |
| Axle 3 mass ($M_{aL}^3$, $M_{aR}^3$/kg) | 325 |
| Suspension stiffness ($K_{uL}^3$, $K_{uR}^3$/kN·m$^{-1}$) | 1969.03 |
| Suspension damping ($C_{uL}^3$, $C_{uR}^3$/kN·s·m$^{-1}$) | 7.18 |
| Tire stiffness ($K_{lL}^3$, $K_{lR}^3$/kN·m$^{-1}$) | 3507.43 |
| Tire damping ($C_{lL}^2$, $C_{lR}^2$/kN·s·m$^{-1}$) | 2 |
| $L_1$/(m) | 1.7 |
| $L_1$/(m) | 2.57 |
| $L_1$/(m) | 1.98 |
| $L_1$/(m) | 2.28 |
| $L_1$/(m) | 2.22 |
| $L_1$/(m) | 2.34 |
| $b$/(m) | 1.1 |

*3.3. Displacement Coupling Method*

Early studies on vehicle–bridge coupling were based on the plane beam and rod model, so the vehicle–bridge coupling equation can be solved by analytical methods or finite element methods. When the bridge adopts a 3D solid element, the vehicle solid element model can be established in the finite element software, and the vehicle–bridge coupling dynamic response can be solved by the method of displacement coupling and transient analysis. Previous studies have shown that the results obtained using the displacement coupling method are highly consistent with the theoretical solution curve that takes all influencing factors into account [44]. Given that the bridge model in this paper is a 3D solid model, and the vehicle model is a spatial vehicle model, the analysis of vehicle–bridge coupling vibration in this study was conducted through the utilization of the displacement coupling method in ANSYS.

The displacement coupling method is implemented through ANSYS transient dynamic analysis. The approach involves applying varying horizontal constraints on the moving mass based on its velocity, and coupling the moving mass with the vertical displacement of the node at the corresponding position. When utilizing the displacement coupling

method for vehicle–bridge coupling dynamic analysis in ANSYS, it is crucial to consider the following issues [44].

(a) Regardless of whether the load step is incremental (KBC = 0) or stepped (KBC = 1), it is recommended to set the substep number to 1 (NSUBST = 1).

(b) The modal damping ratio cannot be specified in the ANSYS complete method for transient dynamic analysis. Instead, the equivalent Rayleigh damping assumption can be utilized, wherein the mass damping coefficient ($\alpha$) and stiffness damping coefficient ($\beta$) are employed. However, this approach results in a spurious damping term ($\alpha M$) during computation, which is not accounted for in the theoretical derivation. Hence, if results with damping are to be compared, only stiffness damping should be taken into consideration.

(c) When using the CP command to couple degrees of freedom, the coupling is linear, so it is not suitable for situations with large deformation. The NLGEOM command can be used to disable large deformation.

(d) The displacement response is largely unaffected by the magnitude of the load step; however, to achieve a more favorable acceleration response, it is advisable to employ a smaller load step.

## 4. Finite Element Method of Typical Damage Characterization

This paper investigates the effects of existing macroscopic damages on SCCBBs, and thus does not include numerical simulations of the damage processes and mechanisms. To analyze the impact of different types of damage on the overall mechanical performance of SCCBBs, the authors selected seven types of damage and defined five grades of damage severity for each type. This resulted in a total of 35 damage scenarios for the numerical analysis models, as presented in Table 3. The seven types of damage are deck breakage, concrete slab stiffness degradation, stud fractures, microcracks in steel beams, diaphragm stiffness degradation, bearing damage, and pier stiffness degradation. This section discusses the methods used to characterize damage for the bridge deck, concrete structure, steel beam structure, bearings, and studs, and also defines formulas for assessing the severity of damage for different types of damage.

**Table 3.** Calculated working conditions.

| Damage Type | Damage Grade |
|---|---|
| deck breakage | I, II, III, IV, V |
| concrete slab stiffness degradation | I, II, III, IV, V |
| stud fracture | I, II, III, IV, V |
| steel beam microcracks | I, II, III, IV, V |
| diaphragm stiffness degradation | I, II, III, IV, V |
| bearing damage | I, II, III, IV, V |
| pier stiffness degradation | I, II, III, IV, V |

### 4.1. Deck Breakage

There are various forms of deck damage, including deck cracks, hugging, ruts, fragmentation, and concrete detachment. These different types of deck damage can lead to irregularities in the deck, which are significant factors affecting the dynamic behavior of vehicle–bridge coupling [47,48]. Thus, this section utilizes the method of simulating deck irregularities to characterize deck damage. In the field of research on vehicle–bridge coupled vibration problems, most scholars consider bridge deck irregularities to be a Gaussian random process with a zero mean value [49,50]. The power spectrum of bridge deck irregularities can be expressed as follows:

$$G_d(n) = G_d(n_0)(n/n_0)^{-w} \tag{13}$$

where $G_d(n_0)$ is the bridge deck irregularity coefficient, which is determined according to the grade of bridge deck irregularity in GB/T7031-2005 [51], "Vehicle vibration—Describing approach for road surface irregularity", as shown in Table 4. $n_0$ is the reference spatial frequency, and the value is 0.1 m$^{-1}$. $w$ is the frequency index, usually 2; $n$ is the spatial frequency (m$^{-1}$).

**Table 4.** Bridge deck irregularity coefficient.

| Deck Grade | Bridge Deck Irregularity Coefficient/$G_d(n_0)$ | | |
| --- | --- | --- | --- |
| | Lower Limit | Geometric Mean | Upper Limit |
| A | 8 | 16 | 32 |
| B | 32 | 64 | 128 |
| C | 128 | 256 | 512 |
| D | 512 | 1024 | 2048 |
| E | 2048 | 4096 | 8192 |

The power spectrum of the bridge deck is simulated using a fast Fourier transform, as depicted in Equation (14).

$$r(x) = \sum_{k=1}^{N} \left( 4G_d(n_0) \left( \frac{2\pi k}{L_c n_0} \right)^{-2} \frac{2\pi}{L_c} \right)^{0.5} \cos\left( \frac{2\pi k n_0}{L_c} + \theta_k \right) \tag{14}$$

where $L_c$ is the length of the deck irregularity. In this paper, the deck irregularity before and after the vehicle moves onto and off the bridge is not considered, but only the level irregularity of six wheels when the vehicle runs on the bridge. $\theta_k$ is an independent set of uniformly distributed random variables obeying $[0, 2\pi]$.

Using the deck irregularity power spectrum method to characterize deck breakage can effectively simulate the level of deck irregularity. However, this method cannot quantitatively simulate the diameter and area of deck breakage. To investigate the sensitivity of deck breakage to the mechanical properties of SCCBBs, this study defines five breakage grades of the deck, I, II, III, IV, and V, based on the five deck grades (A, B, C, D, and E) specified in GB/T7031-2005, "Vehicle vibration—Describing approach for road surface irregularity". The high and low irregularity function values of the five breakage grades of the bridge deck are then derived based on Equations (13) and (14) using the MATLAB software. The high and low irregularity values corresponding to the five breakage grades of the bridge deck are quantitatively characterized by modifying the $G_d(n_0)$ in the program function. The $G_d(n_0)$ values for each breakage grade of the bridge deck are provided in Table 4. The high and low irregularity curves of the five deck breakage grades along the bridge length direction, derived using MATLAB, are shown in Figure 3. Finally, the irregularity function values of different deck breakage grades are stored in a TXT file format in the ANSYS working directory and imported into the ANSYS vehicle–bridge coupling analysis program using the "$Ce$" command to simulate different deck breakage grades.

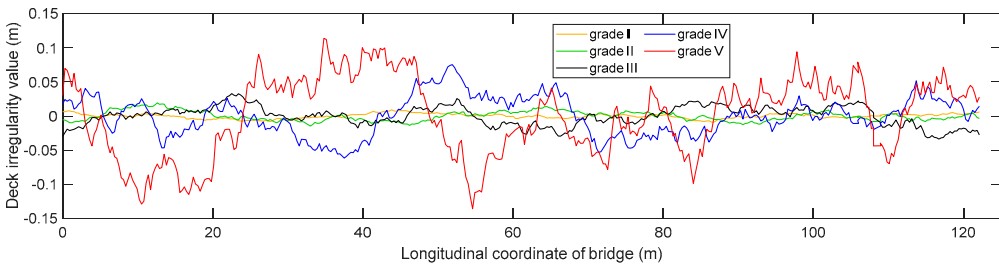

**Figure 3.** The irregularity values of different deck breakage grades.

*4.2. Concrete Structure Damage*

Concrete cracking and other visible damages can cause a decrease in structural stiffness, which is mainly influenced by factors such as the elastic modulus and cross-sectional size [49]. There are various simulation methods available for concrete cracking, including the local stiffness reduction method, separate crack model, and two-dimensional (2D) or 3D crack model. In the local stiffness reduction method, only the elements around the crack undergo stiffness reduction. This reduction should accurately reflect the damage caused by the crack to the bridge structure, which can be represented by the crack position and depth function [52], as illustrated by Equations (15) and (16).

$$E'I' = EI(x)\left(1 - \alpha \cos^2\left(\frac{1}{2}\left(\frac{|x_c - l_c|}{\beta L'/2}\right)^m\right)\right) \tag{15}$$

$$l_c - \beta L'/2 < x_c < l_c + \beta L'/2 \tag{16}$$

where $EI(x)$ represents the stiffness before reduction; $l_c$ is the distance from the center of the crack zone to the left node of the element; $L'$ is the element length; $\alpha$, $\beta$, $m$ are damage parameters, and the range of values refers to the reference [52].

The separate crack model entails creating gaps between neighboring elements during the modeling process, which are determined by the crack's size. This is followed by the addition of bending spring elements to replicate the crack's effect on the structural stiffness. The computation of the bending spring stiffness is demonstrated in Equations (17)–(20) [49].

$$k = 1/c \tag{17}$$

$$c = 6\pi\gamma^2 h f(\gamma) \tag{18}$$

$$\gamma = a/h \tag{19}$$

$$f(\gamma) = 0.6384 - 1.035 + 3.7201\gamma^2 - 5.1773\gamma^3 + 7.553\gamma^4 - 7.332\gamma^5 + 2.4909\gamma^6 \tag{20}$$

where $c$ is the flexibility of the bridge structure; $\gamma$ is the relative depth of the crack; $a$ is the crack depth; $h$ is the beam height.

The 2D or 3D crack model entails the process of refining and removing finite element meshes at the crack location to match its actual size. This technique results in a finite element model that accurately reflects the dimensions of the crack, but it incurs higher computational costs. For situations involving concrete damage and spalling, ANSYS can refine the mesh based on the actual volume of damage or spalling. Additionally, using Boolean operations, 3D solid concrete elements that have been damaged or spalled can be removed [49], as depicted in Figure 4.

*4.3. Steel Structure Damage*

The types of damage to steel beams mainly consist of local instability of the steel plate, residual deformation resulting from plastic deformation, and brittle fracture of the steel caused by low-cycle fatigue loading [53]. The expansion of internal microcracks in steel is the primary cause of damage to steel beams, and numerous microcracks typically exist in the steel beam before a macroscopic fracture occurs, with relatively small dimensions [54]. However, there is currently limited research on the mapping relationship between microcracks in steel beams and the stiffness of steel structures. Hence, this paper employs the local stiffness reduction method outlined in Section 4.2 to simulate microcracks in steel beams. It should be noted that during the process of establishing the finite element model, the material properties should be substituted with the corresponding steel properties before local stiffness reduction is implemented. In the event of a macroscopic fracture occurring

in the steel plate beam, shell or solid elements can be utilized to construct the beam, and a 2D or 3D crack model based on Section 4.2 can be adopted to simulate the fracture, as shown in Figure 4.

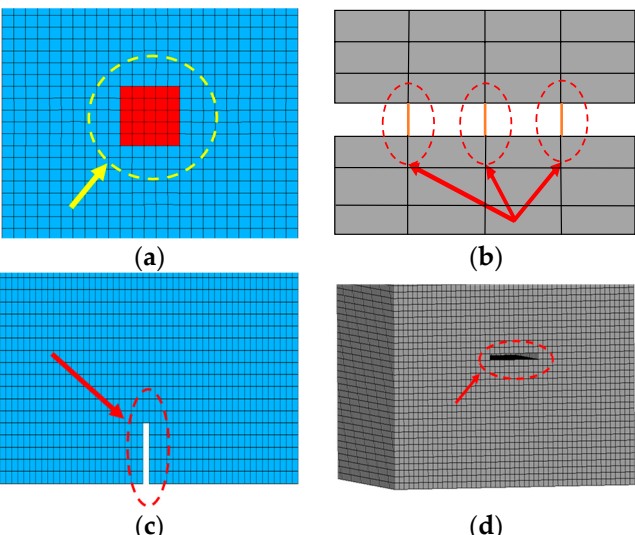

**Figure 4.** Fracture model. (**a**) Local stiffness reduction. (**b**) Separate crack model. (**c**) 2D crack model. (**d**) 3D crack model.

The lateral connection, or transverse diaphragm, between steel beams can be simulated using the 3D rod element LINK8. This element can simulate trusses, links, cables, and springs and can withstand axial tension and compression, but not bending moments [46]. Damage to the lateral connection can be characterized by removing the damaged 3D rod element LINK8 at the damaged location [55]. Additionally, the lateral connection can also be simulated by setting up COMBIN40 combined spring elements in the vertical and horizontal directions [56]. This element offers a range of material settings, including the stiffness ($K1$, $K2$), damping coefficient I, mass ($M$), gap size ($GAP$), and limit sliding force ($FSLIDE$) of the spring element. To simulate damage to the lateral connection, the stiffness and mass material constants of the spring element at the damaged location can be set to zero. If shell or solid elements are used to establish the lateral connection, the damage can be implemented using the local stiffness reduction method in Section 4.2, with attention paid to replacing the steel material properties.

*4.4. Bearing Damage*

(1) Solid bearing model: The utilization of 3D solid elements offers a more accurate depiction of the bearing structure's damage [55]. For instance, to model a typical rubber bearing, the simulation involves implementing the 3D eight-node hyperelastic solid element HYPER58 to represent the rubber layer, based on the Mooney–Rivlin model of continuum mechanics theory. Meanwhile, the steel plate can be replicated through the use of the 3D solid element SOLID45. The bearing is then created by linking the nodes between the rubber and steel plate with the common-node method. In designing and manufacturing rubber bearings, the compressive and shear elastic moduli serve as crucial mechanical performance indicators. Therefore, the simulation of bearing damage can commence with these two parameters. The compressive elastic modulus and shape factor formulas for rubber bearings can be found in the "Highway Bridge Plate Rubber Bearing" (JT/T4-2004) specification, as presented in Equations (21)–(23).

$$E = 5.4 G_e S^2 \tag{21}$$

$$S_1 = \frac{l_{0a}l_{0b}}{2t_1(l_{0a} + l_{0b})} \tag{22}$$

$$S_2 = \frac{d_0}{4t_1} \tag{23}$$

where $E$ represents the compressive elastic modulus; $S_1$ and $S_2$ represent the shape factors of rectangular and circular rubber bearings, respectively; $d_0$ is the diameter of the circular reinforcing steel plate in millimeters; $l_{0a}$ and $l_{0b}$ are the length and width of the rectangular reinforcing steel plate in millimeters, respectively; $t_1$ represents the thickness of the middle rubber layer in millimeters; and $G_e$ represents the shear modulus in megapascals.

Based on Equations (21)–(23), it is evident that the shape factor of the bearing is determined by the planar geometric dimensions and the thickness of the rubber layer. The shape factor ($S$) and shear modulus ($G_e$) have a direct impact on the compressive elastic modulus of the bearing. Therefore, damage to the bearing can be simulated by modifying the planar dimensions and shear modulus of the rubber layer. In ANSYS, modifying the material's elastic modulus is relatively easier compared to directly altering the shear modulus. According to reference [57], a relationship exists between the rubber's elastic modulus $E_0$ and shear modulus $G$ under small strain conditions:

$$G = \frac{E_0}{2(1 + \mu)} \tag{24}$$

where $\mu$ represents Poisson's ratio, which is equal to 0.5 due to the incompressibility of rubber materials. Based on the above analysis, the damage to the bearing can be achieved by changing the planar dimensions and elastic modulus of the rubber material.

(2) Spring bearing model: The hysteresis curve of a plate rubber bearing is typically narrow and elongated, approximating a linear relationship [55]. While ANSYS offers a comprehensive library of elements, it lacks a direct element for isolating rubber bearings. Although the solid bearing model yields high calculation accuracy, it suffers from poor convergence and high computational costs. Therefore, to achieve an approximate simulation, the mechanical model of the bearing can be simplified, and the spring element in ANSYS can be chosen.

Based on existing research results and ANSYS-related literature [46], and after multiple trial calculations and comparisons, this paper simplifies the mechanical model of the isolation rubber bearing to horizontal two-directional nonlinear spring elements, viscous dampers, and vertical linear spring elements [46]. According to this simplification, the vertical stiffness of the bearing can be simulated by linear spring COMBIN14, and the horizontal stiffness can be simulated by nonlinear spring element COMBIN40. COMBIN14 can be set as a 3D axial linear spring through *keyopt 3* of the element, and the vertical stiffness damage of the bearing can be characterized by modifying the spring element constant $K$. The influence of the bilinear strengthening model and viscous damping can be introduced into the COMBIN40 element, and basic parameters such as the pre-yield strength ($K_u$), post-yield strength ($K_d$), yield force ($Q_d$) and damping ratio of the bearing can be set through the real constants of the element. The stiffness damage in both horizontal directions of the bearing can be simulated by modifying the real stiffness constants *K1* and *K2* of the COMBIN40 element [58].

The vertical ($k_z$), transverse ($k_y$), and longitudinal ($k_x$) stiffness formulas of rubber isolation bearings are shown in Equations (25) and (26).

$$k_z = E_e A_e / t_1 \tag{25}$$

$$k_x = k_y = G_e A_e / t_1 \tag{26}$$

where $E_e$ represents the compressive modulus of elasticity (Mpa); $A_e$ represents the bearing area (mm$^2$); $t_1$ represents the thickness of the rubber layer (mm); $G_e$ represents the shear modulus (Mpa).

### 4.5. Stud Damage

The fracturing of studs due to insufficient tensile or shear strength is a common form of damage [59,60]. Once the shear studs fracture, they cease to contribute to the structural force, and the effect of stud fracture can be replicated by reducing the stiffness constant $K$ of the COMBIN14 spring element, as has been done in this study. In ANSYS, the simulation of studs can be achieved using the spring element COMBIN14, and a 3D axial spring can be established by applying *keyopt 3* of the element. The axial and shear stiffness of the studs can be designated by the constant $K$ of the element. The "Specifications for Design and Construction of Highway Steel-Concrete Composite Bridge" (JTG/TD64-01-2015) provide the calculation formulas for the shear and axial stiffness of studs, which are presented in Equations (27) and (28).

$$k_{ss} = 13.0 d_{ss} \sqrt{E_c f_{ck}} \tag{27}$$

$$k_v = E_s A_s / l \tag{28}$$

where $k_{ss}$ is the shear stiffness of the stud connector (N/mm); $d_{ss}$ represents the diameter of the stud connector (mm); $E_c$ represents the elastic modulus of concrete (Mpa); $f_{ck}$ represents the standard compressive strength of concrete (Mpa); $k_v$ represents the axial stiffness of the stud connector; $E_s$ represents the elastic modulus of the stud connector material (Mpa); $A_s$ represents the cross-sectional area of the stud connector rod (mm$^2$); and $l$ represents the length of the stud connector rod (mm).

### 4.6. Classification of Damage Grade

The classification method of the damage grade of deck breakage has been explained in Section 4.1 of the article, and will not be repeated here. Considering the presence of numerous cracks in deteriorated bridge structures, with their sizes being negligible when compared to the overall size of the structure, accurately characterizing their numbers using fine-meshed 2D or 3D crack models is difficult. Moreover, a refined mesh may result in convergence problems. Therefore, based on the available research [61–63], this paper assumes that cracks are evenly distributed on the concrete slab, steel beams, diaphragm, and bridge piers, and modifies the overall elastic modulus of the cracked structure using the method of local stiffness reduction (this can be achieved in ANSYS using the MP command; the replacement of material properties for concrete and steel should be taken into account) [64]. The elastic moduli for the five damage gradient grades are presented in Equation (29).

$$E_i = [1 - 0.2(i-1)]E, \ i = (1, 2, 3, 4, 5) \tag{29}$$

where $i$ is the damage grade; $E_i$ is the overall elastic modulus of the corresponding structure when it is damaged at grade $i$; $E$ is the initial elastic modulus of the material used in the damaged structure.

The stud is modeled using the 3D spring element COMBIN14, while the bearing is simulated using a combination of the COMBIN14 and COMBIN40 elements. The various damage grades of the stud and bearing can be achieved by modifying the element's real constant. Equation (30) provides the relationship between the real constant $K$ and the five damage gradient grades for both the bearing and the stud. Similarly, the transverse and longitudinal stiffness constants $K1$ and $K2$ for the bearing are also derived.

$$K_i = [1 - 0.2(i-1)]K, \ i = (1, 2, 3, 4, 5) \tag{30}$$

where $i$ is the damage grade; $K_i$ is the spring stiffness constant corresponding to class $i$ damage; $K$ is the spring stiffness constant when the bearing or stud is not damaged.

## 5. Damage Sensitivity Analysis Method

### 5.1. Sensitivity Analysis Method Based on ET

Compared with traditional decision trees, the extremely randomized forests (ET) algorithm has a stronger generalization ability and stronger explanatory ability compared with neural networks and support vector machines [65].

In the field of machine learning research, researchers have already employed machine-learning-based sensitivity analysis methods to assess the potential damage caused by earthquake ground motion [34,36,37]. Firstly, the intensity index value of a given ground motion sample is calculated and input into the ET model. Then, it is classified step by step according to the given intensity index value and finally falls into the leaf node. The average value of the output results of each decision leaf node is the evaluation value of the ground motion damage index, which can be used to evaluate the potential damage potential of ground motion. This method quantifies the influence of input parameters using sensitivity coefficients based on ET, as illustrated in Figure 5. This approach can be utilized to calculate the sensitivity coefficient $V_t$ of any damage indicator to strength indicators, and the precision of $V_t$ improves with the complexity of the ET network. Assuming that a strength indicator is *Imx*, the specific calculation steps for its sensitivity coefficient $V_t$ are as follows [35]:

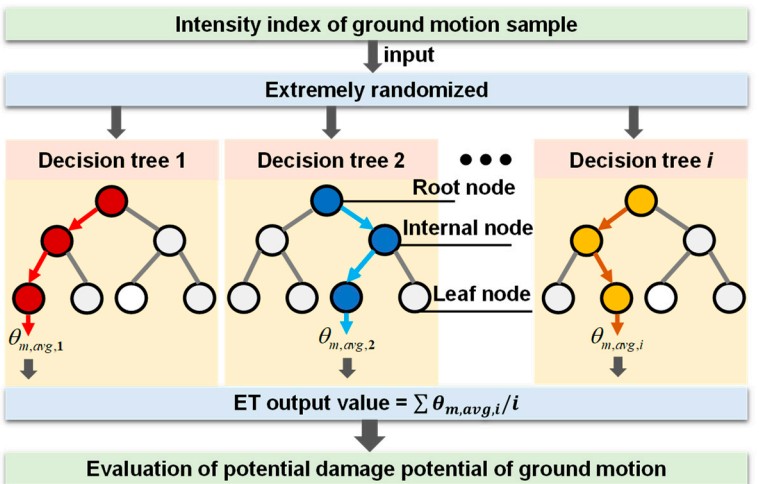

**Figure 5.** ET network structure and failure potential assessment process.

Step 1: Compute the sensitivity coefficient $V_{Imx,k,ij}$ of the strength indicator *Imx* at node $v_{ij}$ in decision tree $k$, following Equation (31).

$$V_{IMx,k,ij} = H(X_{ij}) - G_{ij} \tag{31}$$

where $X_{ij}$ represents all data sets before the split at node $v_{ij}$; $H(X)$ is the impurity function; $G_{ij}$ is the weighted impurity.

Step 2: Sum the sensitivity coefficients of all nodes, and use Equation (32) to compute the sensitivity coefficient $V_{Imx,k}$ of *Imx* in decision tree $k$.

$$V_{IMx,k} = \sum_{m \in M} V_{IMx,k,m} \tag{32}$$

where $M$ is the set of all nodes in decision tree $k$ that use *Imx* as the splitting feature, and $V_{Imx,k,m}$ is the sensitivity coefficient of strength indicator *Imx* at node $m$ in set $M$, which can be calculated using Equation (31).

Step 3: Sum the sensitivity coefficients of all decision trees, and use Equation (33) to compute the sensitivity coefficient $V_{Imx}$ of the importance measure *Imx* in the ET forest.

$$V_{IMx} = \sum_{k=1}^{n} V_{IMx,k} \tag{33}$$

Step 4: Utilizing the sensitivity coefficients of all strength indicators, normalize the sensitivity coefficient of *Imx* to obtain the global sensitivity coefficient $V_{t,Imx}$, through Equation (34).

$$V_{t,IMx} = V_{IMx} / \sum_{h \in H} V_{IMh} \tag{34}$$

where $H$ is the set of all strength indexes involved in calculation; $V_{Imh}$ is the sensitivity coefficient of index *Imh* in set $H$ in ET.

### 5.2. Damage Sensitivity Analysis Method for SCCBBs

To quantify the sensitivity of SCCBBs to various types of damage and enable a quantitative evaluation of the damage grade, this section employs the sensitivity analysis technique of ET and introduces a damage sensitivity analysis method for SCCBBs. The approach incorporates the damage type, damage grade, and multiple static and dynamic indicators to determine the sensitivity grades of different damage types to the overall mechanical behavior of the bridge structure. This is achieved by solving the size of the sensitivity impact factor, as depicted in Figure 6.

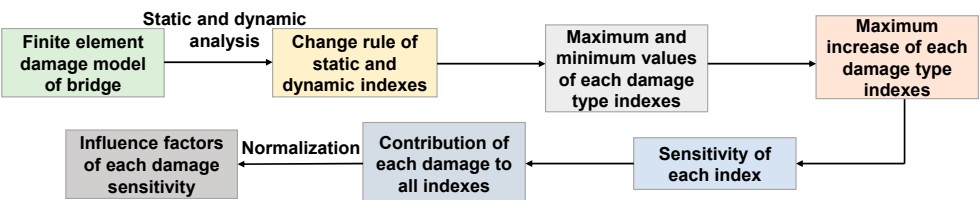

**Figure 6.** Technology roadmap.

To facilitate research, a fundamental assumption of the linear accumulation of damage is adopted. During the calculation process, the impact of each type of damage on the static and dynamic indicators of the bridge is assumed to be linearly related. This means that the effects of multiple types of damage on the bridge structure can be directly superimposed. To enhance readers' comprehension, a graph illustrating the relationship between damage and indicators (shown in Figure 7) is presented. Additionally, four calculation and analysis formulas (Equations (35)–(41)) are proposed to further elaborate on the method of damage sensitivity analysis. The specific analysis steps are outlined below.

Step 1: Construct numerical analysis models for SCCBBs with various types and grades of damage.

Step 2: Perform static and dynamic analyses on the finite element models for each type of damage at each damage grade, and refer to Figure 7 for the relationships between damage and indicators.

Step 3: Compute the maximum and minimum values of static and dynamic indicators for each damage type.

$$\begin{cases} X_{i,\max}^k = \max\left\{ N_{i,1}^k, \cdots N_{i,j}^k \right\} \\ X_{i,\min}^k = \min\left\{ N_{i,1}^k, \cdots N_{i,j}^k \right\} \end{cases} \tag{35}$$

where $X_{i,max}^k$ and $X_{i,\min}^k$ are the maximum and minimum values of the *k*th index for type *i* damage at different damage grades, respectively.

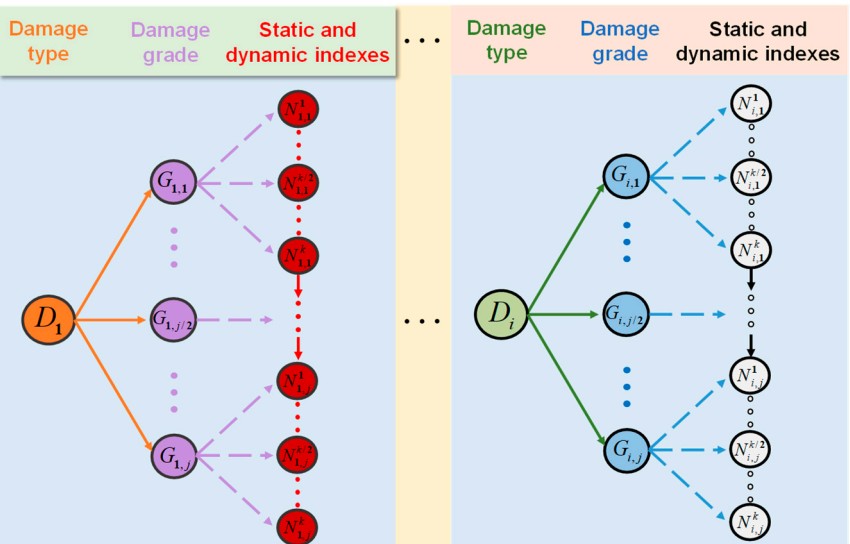

**Figure 7.** Damage–index relationship. In the figure, $D_i$ is the class $i$ injury, $G_{i,j}$ is the class $i$ injury of grade $j$, and $N_{i,j}^k$ is the $k$th indexes of the class $i$ injury of grade $j$.

Step 4: Determine the maximum increase $A_{i,max}^k$ for each static and dynamic indicator of each damage type.

$$A_{i,\max}^k = \frac{X_{i,\max}^k - X_{i,\min}^k}{X_{i,\min}^k} \tag{36}$$

where $A_{i,max}^k$ is the maximum increase value of the $k$th index of type $i$ damage.

Step 5: Solve the mechanical index sensitivity $V_i^k$ for each damage type. The index sensitivity reflects the impact of the damage type on a specific index, as demonstrated in Equations (37) and (38).

$$A_{\max}^k = \sum_{i=1}^{n} A_{i,\max}^k \tag{37}$$

$$V_i^k = \frac{A_{i,\max}^k}{A_{\max}^k} \tag{38}$$

where $V_i^k$ is the $k$th index sensitivity of type $i$ damage. $N$ is the number of damage types. $A_{max}^k$ represents the sum of the effects of $n$ types of damage on the maximum increase in mechanical index $k$.

Step 6: Calculate the contribution of each damage type to the impact on all indices, and sum them according to the damage type, as presented in Equation (39).

$$V_i = V_i^1 + V_i^2 + \cdots + V_i^k \tag{39}$$

where $V_i$ is the contribution of the influence of type $i$ damage on all mechanical indexes.

Step 7: Normalize the sensitivity impact factor $\mu_i$ for each damage type.

$$V = \sum_{i=1}^{n} V_i \tag{40}$$

$$\mu_i = \frac{V_i}{V} \tag{41}$$

## 6. Damage Sensitivity Analysis of the SCCBB

Using the vehicle–SCCBB interaction model outlined in Section 3 and the seven damage types established in Section 4, we developed a numerical simulation model of the

SCCBB with five damage grades and a 12-degree-of-freedom three-axis vehicle model. In order to evaluate the sensitivity of the seven damage types, Section 5.2's SCCBB damage sensitivity analysis method was employed, using eight selected static and dynamic indicators. These indicators include the maximum deflection of the side span, the maximum stress of the structure, the maximum displacement of the pier top, vertical vibration displacement (VVD), vertical vibration acceleration (VVA) at the middle of the side span, VVD and VVA at the middle of the midspan, and the first-order modal frequency.

To ensure the accuracy and reliability of the analysis results, we considered the worst-case scenario by incorporating both static and dynamic loads. The static load was determined based on the lane load specified in the "General Specifications for Design of Highway Bridges and Culverts" (JTG D60-2015), with the loading position arranged for a two-lane road. For the dynamic load, we utilized the three-axis vehicle model described in Section 3.2. As noted by Li et al. [47], the vertical vibration displacement (VVD) at the side span of a steel–concrete composite continuous beam bridge gradually increases with vehicle speed, assuming a constant vehicle weight. Therefore, we selected a design speed of 100 km/h for the vehicle speed in this study.

### 6.1. Static and Dynamic Results

#### 6.1.1. Static Results

To investigate the impact of various damage types on the static characteristics of a steel–concrete composite continuous beam bridge, we selected three static indicators for analysis: the maximum deflection of the side span, the maximum stress of the structure, and the maximum displacement of the pier top. Figure 8 presents the analysis results, with the *X*-axis indicating the seven damage types, the *Y*-axis representing the damage grades, and the *Z*-axis reflecting the three static indicators.

Based on Figure 8, microcracks in the steel beam result in a maximum deflection of 86.7 mm at the midpoint of the side span, while other damage types display relatively flat deflection surfaces ranging from 30 to 56 mm. Under static lane loads, the highest stress in the bridge structure appears at the bearing, with a maximum stress of 254 Mpa when the stiffness degradation of the concrete slab reaches grade V. The maximum stress from bearing damage decreases as the damage grade rises. This trend is similarly observed when simulating the impacts of diaphragm stiffness degradation and steel beam microcracks using the local stiffness reduction method. Therefore, scientific reduction of the diaphragm and steel beam stiffness may help to alleviate stress concentration at the bearing.

The maximum displacement at the top of the pier due to concrete slab stiffness degradation is 1.78 mm. The corresponding range surfaces for deck breakage, diaphragm stiffness degradation, stud fracture, and bearing damage exhibit relatively flat changes and have minimal impact on the displacement at the top of the pier.

#### 6.1.2. Dynamic Results

To examine the effects of various damage types on the dynamic characteristics of a steel–concrete composite continuous beam bridge, we selected five dynamic indicators for analysis: VVD, VVA, and the first-order modal frequency at the middle of the side span and midspan. Figures 9 and 10 illustrate the changes in VVD at the middle of the side span and midspan, respectively, with the grade of damage under different damage types. Additionally, Table 5 presents the values of VVD at the middle of the spans under different types and grades of damage.

Observations from Figures 9 and 10, as well as Table 5, indicate that as the degree of damage to the bridge increases, the VVD curves become more erratic. Furthermore, the maximum VVD values at the side span and midspan are 9.80 mm and 9.59 mm, respectively, with maximum increases of 68.5% and 67.3%. The fluctuations in the VVD amplitude are also evident, as concrete stiffness degradation and steel beam microcracks have a significant impact. The maximum VVD values at the side span and midspan are 9.80 mm, 10.10 mm, 62.05 mm, and 72.61 mm, with maximum increases of 67.8%, 962.1%, 62.0%, and 1064.7%,

respectively. Although the displacement amplitude difference between the no-damage state and grade II damage of the cross-diaphragm stiffness is significant, there is no significant change in displacement amplitude as the grade of damage increases. The maximum VVD values at the side span and midspan are 18.53 mm and 20.06 mm, respectively, with maximum increases of 217.2% and 221.7%.

Conversely, the VVD amplitude for three types of damage stud fracture, bearing damage, and pier stiffness degradation remains largely unaffected by the grade of damage. The displacement curves under varying grades of damage demonstrate a high degree of alignment and exhibit a gentle trend. Maximum VVD values at the middle of the side span and midspan are both less than 7.5 mm, with maximum increases within 18.0%.

Figures 11 and 12 depict the variations in VVA at the midspan and side span, respectively, for different damage grades under varied damage effects. Moreover, Table 6 presents the VVA values at the middle span for diverse damage types and grades. As is evident from Figures 11 and 12 and Table 6, the grade of deck breakage exerts the most pronounced impact on the VVA fluctuation range at the midspan. Notably, the fluctuation range of VVA at the side span and midspan is $-2.83.2$ m/s$^2$ and $-2.22.4$ m/s$^2$, respectively, while the maximum acceleration increase among different damage grades is 1483.8% and 1109.1%, correspondingly.

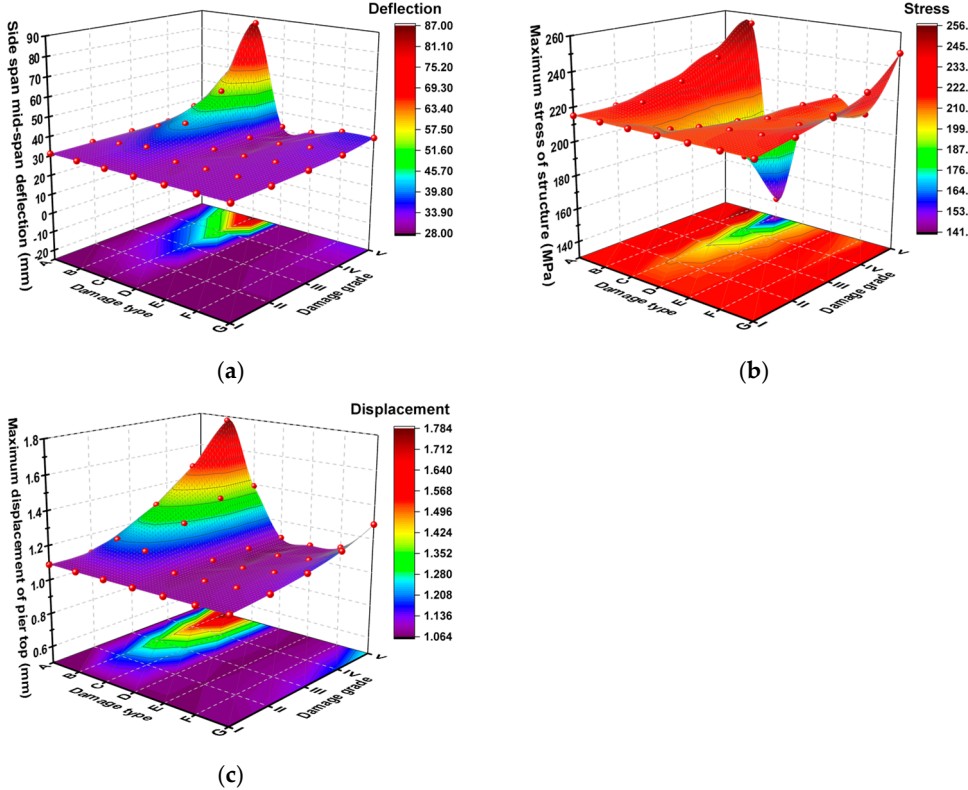

**Figure 8.** Influence of different damages on static characteristics. A is deck breakage, B is concrete slab stiffness degradation, C is steel beam microcrack, D is diaphragm stiffness degradation, E is stud fracture, F is bearing damage, and G is pier stiffness degradation. I~V are the damage grades. (**a**) The deflection in the middle of the side span (mm). (**b**) The maximum stress of structure (Mpa). (**c**) The maximum displacement of the top of the pier (mm).

**Table 5.** The value of midspan VVD under different damage types and grades.

| Position | Damage Type | Grade I | Grade II | Grade III | Grade IV | Grade V | $A_{max}$ |
|---|---|---|---|---|---|---|---|
| | A | 5.8425 | 5.8152 | 6.4920 | 8.0525 | 9.7959 | 68.5% |
| | B | 5.8425 | 6.5072 | 7.0413 | 7.9242 | 9.8059 | 67.8% |
| | C | 5.8425 | 22.5625 | 27.7893 | 37.3555 | 62.0528 | 962.1% |
| Side span | D | 5.8425 | 18.3707 | 18.4050 | 18.4557 | 18.5342 | 217.2% |
| | E | 5.8425 | 5.8825 | 6.2288 | 6.2529 | 6.3885 | 9.4% |
| | F | 5.8425 | 6.2199 | 6.3282 | 6.4964 | 6.8514 | 17.3% |
| | G | 5.8425 | 5.8955 | 5.9731 | 6.1017 | 6.4008 | 9.6% |
| | A | 6.2348 | 5.7332 | 6.5928 | 8.4977 | 9.5906 | 67.3% |
| | B | 6.2348 | 7.0186 | 7.5363 | 8.3520 | 10.1021 | 62.0% |
| | C | 6.2348 | 24.8421 | 30.7657 | 41.9277 | 72.6145 | 1064.7% |
| Midspan | D | 6.2348 | 19.9038 | 19.9365 | 19.9817 | 20.0574 | 221.7% |
| | E | 6.2348 | 6.2859 | 6.7253 | 6.7604 | 6.8296 | 9.5% |
| | F | 6.2348 | 6.7151 | 6.8137 | 6.9702 | 7.3134 | 17.3% |
| | G | 6.2348 | 6.2810 | 6.3455 | 6.4504 | 6.6843 | 7.2% |

Note: A is deck breakage, B is concrete slab stiffness degradation, C is steel beam microcrack, D is diaphragm stiffness degradation, E is stud fracture, F is bearing damage, and G is pier stiffness degradation. $A_{max}$ is the maximum increase.

**Table 6.** The VVA values at the middle span under different damage types and grades.

| Position | Damage Type | Grade I | Grade II | Grade III | Grade IV | Grade V | $A_{max}$ |
|---|---|---|---|---|---|---|---|
| | A | 0.2019 | 0.3676 | 0.6801 | 1.9206 | 3.1980 | 1483.8% |
| | B | 0.2019 | 0.2467 | 0.2989 | 0.3947 | 0.6850 | 239.3% |
| | C | 0.2019 | 0.9208 | 1.0249 | 1.1923 | 1.5644 | 674.7% |
| Side span | D | 0.2019 | 0.7253 | 0.7312 | 0.7288 | 0.7445 | 268.7% |
| | E | 0.2019 | 0.2415 | 0.3047 | 0.2477 | 0.4137 | 104.9% |
| | F | 0.2019 | 0.2168 | 0.2207 | 0.2511 | 0.3777 | 87.1% |
| | G | 0.2019 | 0.2030 | 0.2093 | 0.2153 | 0.2294 | 13.6% |
| | A | 0.1921 | 0.4136 | 0.8613 | 1.1182 | 2.3231 | 1109.1% |
| | B | 0.1921 | 0.2511 | 0.3077 | 0.3978 | 0.6845 | 256.3% |
| | C | 0.1912 | 0.9195 | 1.0492 | 1.3112 | 1.9563 | 918.2% |
| Midspan | D | 0.1912 | 0.7636 | 0.7662 | 0.7701 | 0.7720 | 301.8% |
| | E | 0.1921 | 0.2268 | 0.2601 | 0.2492 | 0.2533 | 35.4% |
| | F | 0.1912 | 0.2172 | 0.2177 | 0.2169 | 0.2258 | 17.5% |
| | G | 0.1912 | 0.1916 | 0.1920 | 0.1921 | 0.1923 | 0.4% |

Note: A is deck breakage, B is concrete slab stiffness degradation, C is steel beam microcrack, D is diaphragm stiffness degradation, E is stud fracture, F is bearing damage, and G is pier stiffness degradation. $A_{max}$ is the maximum increase.

Concerning the degradation of concrete slab stiffness, the topmost fluctuation range of VVA at the side span and midspan is $-0.30.7$ m/s$^2$ and $-0.250.7$ m/s$^2$, respectively, exhibiting a maximum increase of 239.3% and 256.3%, respectively. Concerning transverse diaphragm stiffness damage, there exists a notable dissimilarity in the VVA amplitude at the midspan between grade I (undamaged state) and grade II. However, no evident change occurs in the fluctuation range of the acceleration amplitude with an increase in damage grade beyond grade II.

In the case of stud fracture, bearing damage, and pier stiffness degradation, the acceleration curves for various damage grades display a close match, and the fluctuation ranges are minimal. At the side span and midspan, the highest fluctuation ranges of the acceleration amplitude are $-0.180.42$ m/s$^2$, $-0.40.25$ m/s$^2$, $-0.230.23$ m/s$^2$, $-0.050.27$ m/s$^2$, $-0.110.23$ m/s$^2$, and $-0.050.20$ m/s$^2$, respectively, with the maximum increase being less than 105.0%.

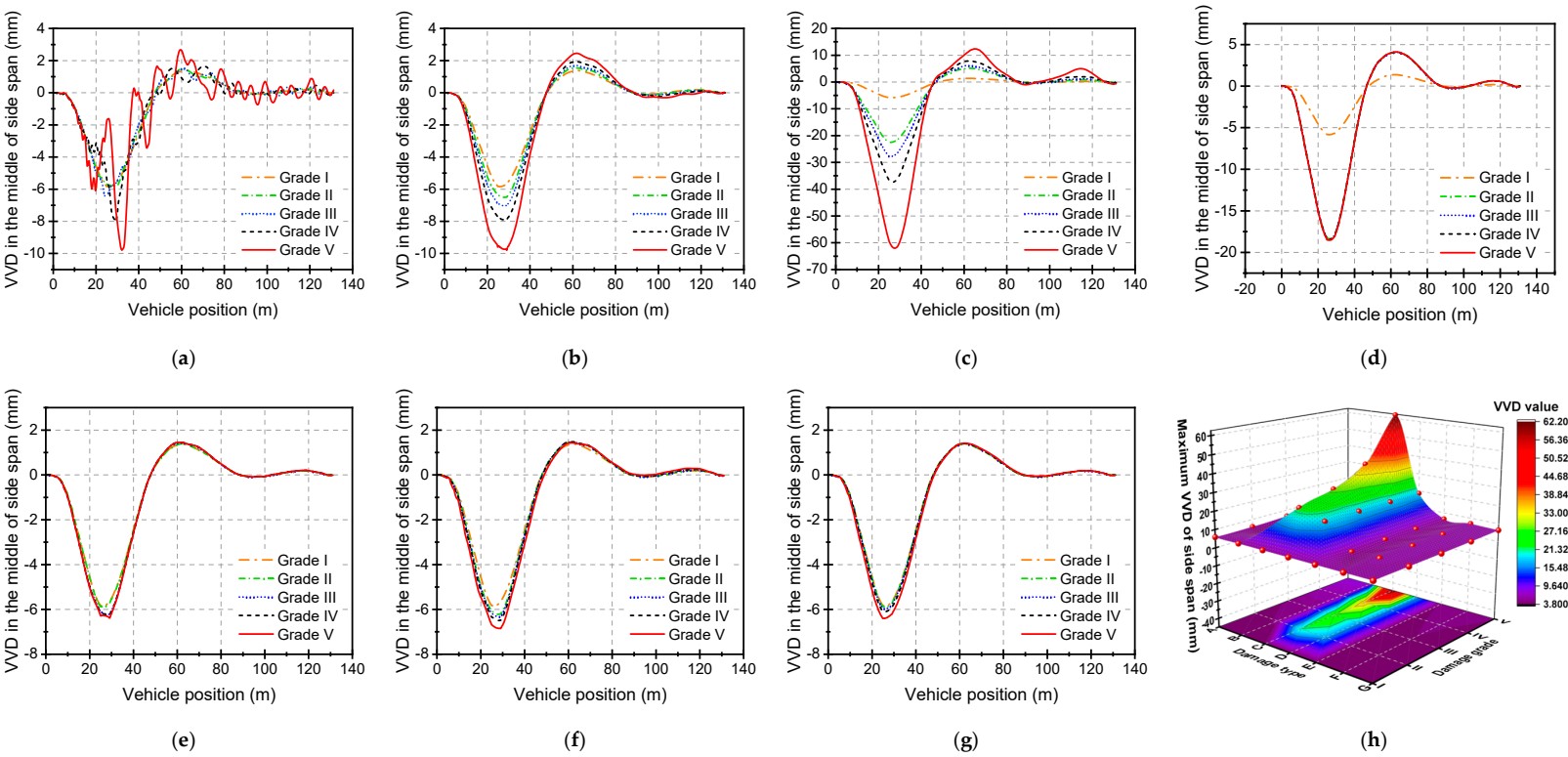

**Figure 9.** VVD of side span under different damage effects. (**a**) VVD of side span under deck breakage (mm). (**b**) VVD of side span under concrete slab stiffness degradation (mm). (**c**) VVD of side span under steel beam microcrack (mm). (**d**) VVD of side span under diaphragm stiffness degradation (mm). (**e**) VVD of side span under stud fracture (mm). (**f**) VVD of side span under bearing damage (mm). (**g**) VVD of side span under pier stiffness degradation (mm). (**h**) Peak value of side span VVD under different damage effects (mm). A is deck breakage, B is concrete slab stiffness degradation, C is steel beam microcrack, D is diaphragm stiffness degradation, E is stud fracture, F is bearing damage, and G is pier stiffness degradation. I~V are the damage grades.

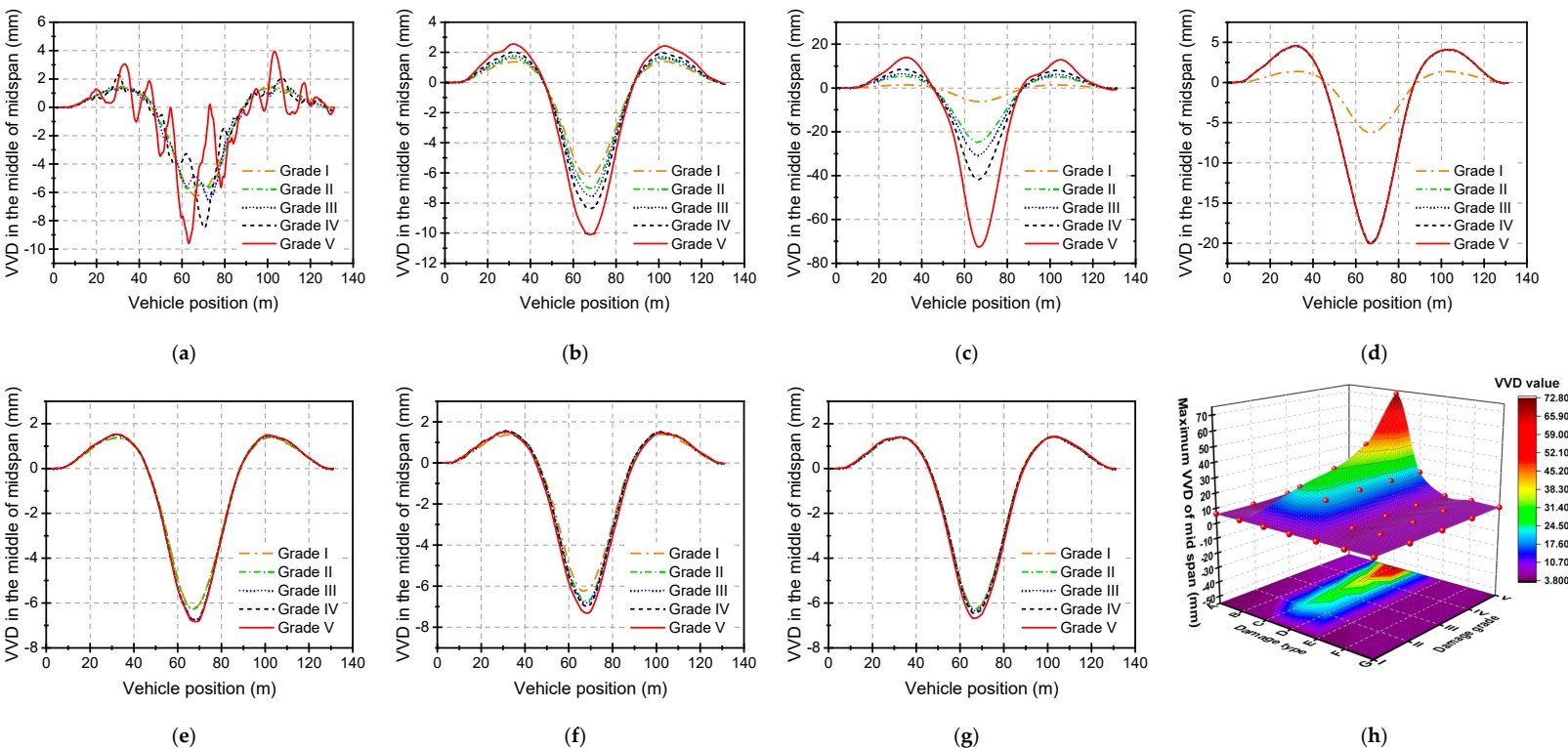

**Figure 10.** VVD of midspan under different damage effects. (**a**) VVD of midspan under deck breakage (mm). (**b**) VVD of midspan under concrete slab stiffness degradation (mm). (**c**) VVD of midspan under steel beam microcrack (mm). (**d**) VVD of midspan under diaphragm stiffness degradation (mm). (**e**) VVD of midspan under stud fracture (mm). (**f**) VVD of midspan under bearing damage (mm). (**g**) VVD of midspan under pier stiffness degradation (mm). (**h**) Peak value of midspan VVD under different damage effects (mm). A is deck breakage, B is concrete slab stiffness degradation, C is steel beam microcrack, D is diaphragm stiffness degradation, E is stud fracture, F is bearing damage, and G is pier stiffness degradation. I~V are the damage grades.

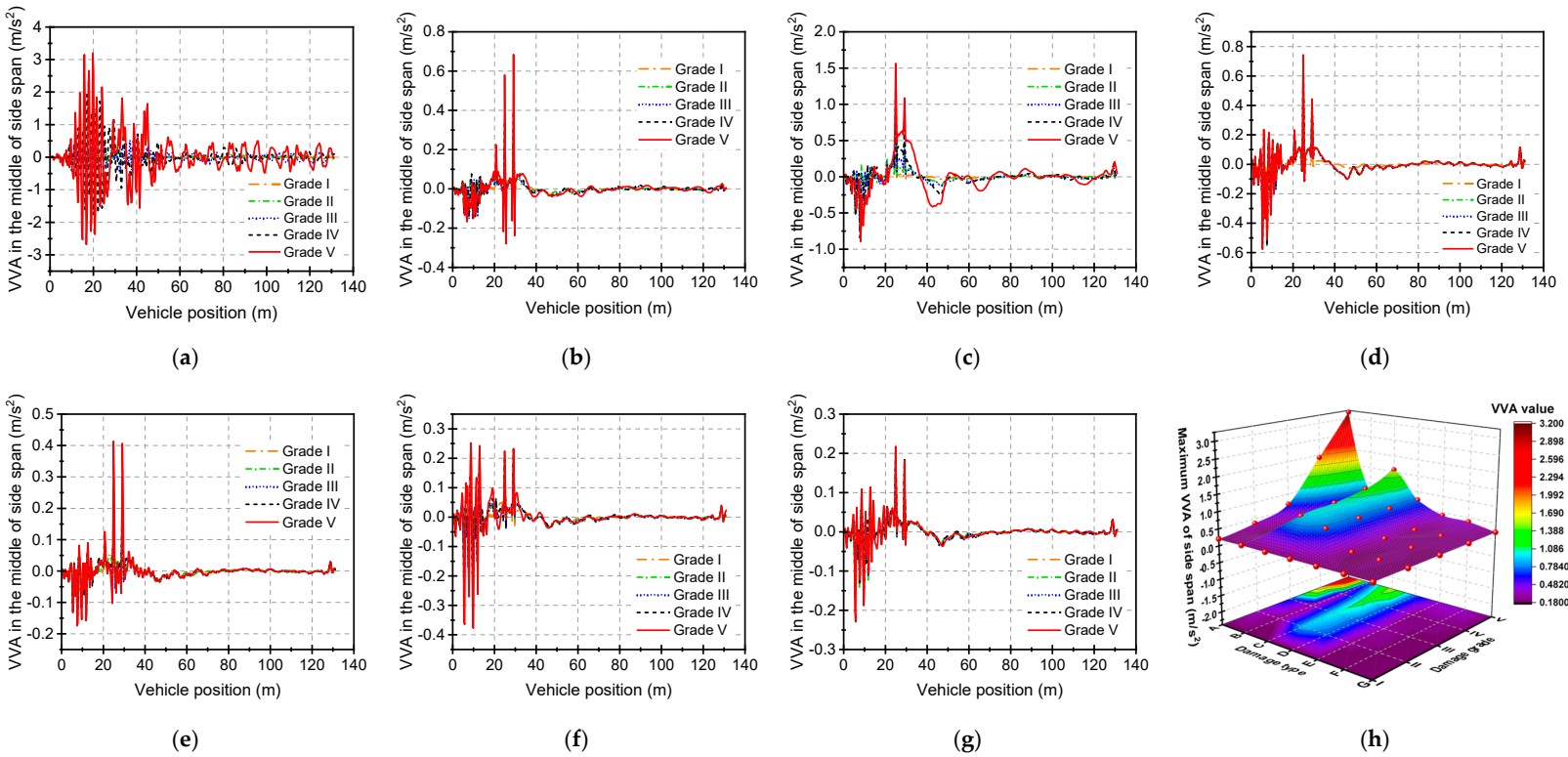

**Figure 11.** VVA of side span under different damage effects. (**a**) VVA of side span under deck breakage (m/s²). (**b**) VVA of side span under concrete slab stiffness degradation (m/s²). (**c**) VVA of side span under steel beam microcrack (m/s²). (**d**) VVA of side span under diaphragm stiffness degradation (m/s²). (**e**) VVA of side span under stud fracture (m/s²). (**f**) VVA of side span under bearing damage (m/s²). (**g**) VVA of side span under pier stiffness degradation (m/s²). (**h**) Peak value of side span VVA under different damage effects (m/s²). A is deck breakage, B is concrete slab stiffness degradation, C is steel beam microcrack, D is diaphragm stiffness degradation, E is stud fracture, F is bearing damage, and G is pier stiffness degradation. I~V are the damage grades.

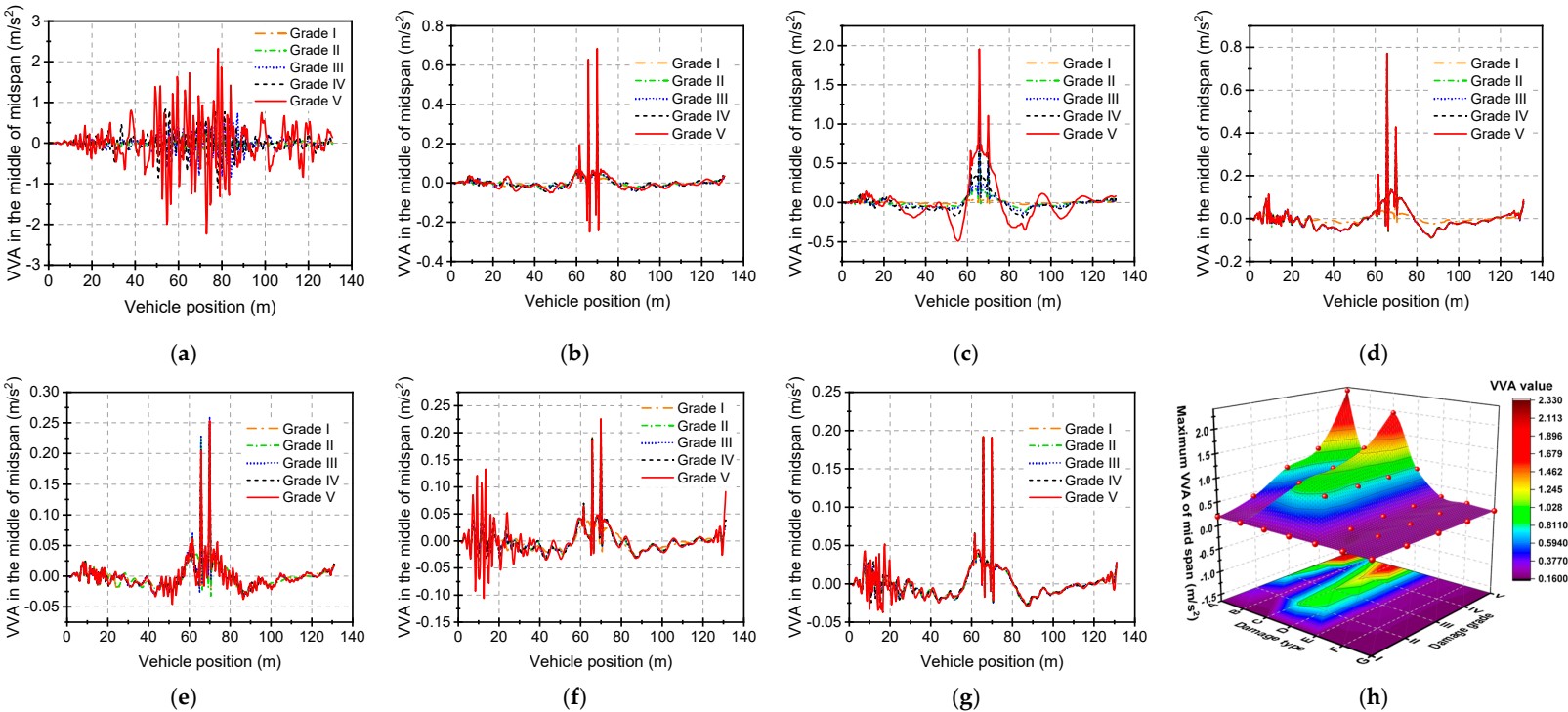

**Figure 12.** VVA of midspan under different damage effects. (**a**) VVA of midspan under deck breakage (m/s$^2$). (**b**) VVA of midspan under concrete slab stiffness degradation (m/s$^2$). (**c**) VVA of midspan under steel beam microcrack (m/s$^2$). (**d**) VVA of midspan under diaphragm stiffness degradation (m/s$^2$). (**e**) VVA of midspan under stud fracture (m/s$^2$). (**f**) VVA of midspan under bearing damage (m/s$^2$). (**g**) VVA of midspan under pier stiffness degradation (m/s$^2$). (**h**) Peak value of midspan VVA under different damage effects (m/s$^2$). A is deck breakage, B is concrete slab stiffness degradation, C is steel beam microcrack, D is diaphragm stiffness degradation, E is stud fracture, F is bearing damage, and G is pier stiffness degradation. I~V are the damage grades.

The frequency index is a vital parameter that reflects the overall structural behavior of a bridge. Damage can cause similar frequency changes in different locations of the structure, particularly in symmetric structures. Figure 13 provides a summary of the first-order modal frequency outcomes corresponding to each damage type under different damage grades. As depicted in Figure 13, for each damage type, the slope of the corresponding curved surface decreases most significantly as the damage grade increases for steel beam microcracks, with a minimum frequency of 1.63 Hz at damage grade V. The slope of the corresponding curved surface for concrete slab stiffness degradation is second only to that of steel beam microcracks, with a minimum frequency of 2.3 Hz. In contrast, the slope of the corresponding curved surface for bearing damage and pier stiffness degradation initially shows no significant changes and only exhibits a slight decrease when the damage grade reaches grade IV or V, with minimum frequencies of 2.70 Hz and 2.72 Hz, respectively. The minimum first-order modal frequencies corresponding to deck breakage, diaphragm stiffness degradation, and stud fracture are 2.85 Hz, 2.83 Hz, and 2.83 Hz, respectively, with the corresponding curved surfaces being almost flat. The frequency values show no significant changes with increasing damage grades and remain around 2.84 Hz.

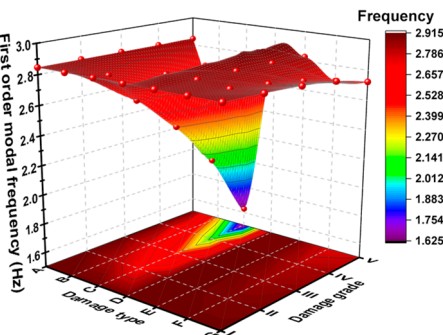

**Figure 13.** First-order modal frequency. A is deck breakage, B is concrete slab stiffness degradation, C is steel beam microcrack, D is diaphragm stiffness degradation, E is stud fracture, F is bearing damage, and G is pier stiffness degradation. I~V are the damage grades.

## 6.2. Index Maximum Value and Increase Analysis

By integrating Equation (35) from Section 5.2 with the findings from the static–dynamic indicators in Section 6.1, we can establish the upper and lower limits of the eight static–dynamic indicators for each type of damage over five damage grades. In order to compute the greatest possible increase in the eight static–dynamic indicators for each damage type, we utilize Equation (36), and we present the outcomes in Table 7.

Table 7 illustrates that the impact of various damage types on the maximum increase in the eight static–dynamic indicators differs. The breakage of the deck has the most notable effect on the maximum values of VVA at the midspan and side span, with maximum increases of 1483.8% and 1109.1%, respectively. Deck breakage also affects VVD at the midspan and side span, but with smaller maximum increases of 68.5% and 67.3%, respectively. The remaining static–dynamic indicators remain unaffected by deck breakage. Concrete slab stiffness degradation has a relatively significant impact on the maximum values of VVD at the midspan and side span, with maximum increases of 239.3% and 256.3%, respectively. Other static–dynamic indicators are also affected by concrete slab stiffness degradation, but with maximum increases within 80.0%. After the steel beam cracks, the maximum values of VVD at the midspan and side span demonstrate the most significant changes, with maximum increases of 962.1% and 1064.7%, respectively. The maximum values of VVA at the midspan and side span also exhibit significant changes, with maximum increases of 674.7% and 918.2%, respectively. The maximum increase in the middle of the side span deflection is 179.1%. The other static–dynamic indicators are less impacted by steel beam microcracks, with maximum increases within 75.0%.

The degradation of the diaphragm stiffness only affects the maximum values of VVD and VVA in the middle of the spans, with the maximum increase fluctuating at around 200.0% to 300.0%. The maximum increases in other static–dynamic indicators, such as the maximum deflection of the side span, maximum stress in the structure, maximum displacement of pier tops, and first-order modal frequency, are all within 7.0%, and their impacts can be disregarded. The impact of stud fracture, bearing damage, and pier stiffness degradation on static–dynamic indicators is relatively small. Stud fracture and bearing damage only have a certain effect on the maximum VVA of the side span, while the maximum increase in the other static and dynamic indices remains within 105.0%.

**Table 7.** Maximum value and maximum increase in static and dynamic indexes.

|  |  | A | B | C | D | E | F | G |
|---|---|---|---|---|---|---|---|---|
| N1 | $X_{max}^1$ | 0.0311 | 0.0557 | 0.0867 | 0.0319 | 0.0319 | 0.0358 | 0.0359 |
|  | $X_{min}^1$ | 0.0311 | 0.0311 | 0.0311 | 0.0311 | 0.0311 | 0.0311 | 0.0311 |
|  | $A_{max}^1$ | 0.0% | 79.1% | 179.1% | 2.7% | 2.7% | 15.3% | 15.6% |
| N2 | $X_{max}^2$ | 215 | 254 | 215 | 215 | 217 | 216 | 248 |
|  | $X_{min}^2$ | 215 | 215 | 142 | 201 | 215 | 209 | 215 |
|  | $A_{max}^2$ | 0.0% | 18.1% | 51.4% | 7.0% | 0.9% | 3.4% | 15.4% |
| N3 | $X_{max}^3$ | 0.0011 | 0.0018 | 0.0014 | 0.0011 | 0.0011 | 0.0011 | 0.0013 |
|  | $X_{min}^3$ | 0.0011 | 0.0011 | 0.0011 | 0.0011 | 0.0011 | 0.0011 | 0.0011 |
|  | $A_{max}^3$ | 0.0% | 63.0% | 25.6% | 0.5% | 0.5% | 2.5% | 18.1% |
| N4 | $X_{max}^4$ | 0.0098 | 0.0098 | 0.0621 | 0.0185 | 0.0064 | 0.0069 | 0.0064 |
|  | $X_{min}^4$ | 0.0058 | 0.0058 | 0.0058 | 0.0058 | 0.0058 | 0.0058 | 0.0058 |
|  | $A_{max}^4$ | 68.5% | 67.9% | 962.1% | 217.2% | 9.4% | 17.3% | 9.6% |
| N5 | $X_{max}^5$ | 3.1980 | 0.6850 | 1.5644 | 0.7445 | 0.4137 | 0.3777 | 0.2294 |
|  | $X_{min}^5$ | 0.2019 | 0.2019 | 0.2019 | 0.2019 | 0.2019 | 0.2019 | 0.2019 |
|  | $A_{max}^5$ | 1483.8% | 239.3% | 674.7% | 268.7% | 104.9% | 87.1% | 13.6% |
| N6 | $X_{max}^6$ | 0.0096 | 0.0101 | 0.0726 | 0.0201 | 0.0068 | 0.0073 | 0.0067 |
|  | $X_{min}^6$ | 0.0057 | 0.0062 | 0.0062 | 0.0062 | 0.0062 | 0.0062 | 0.0062 |
|  | $A_{max}^6$ | 67.3% | 62.0% | 1064.7% | 221.7% | 9.5% | 17.3% | 7.2% |
| N7 | $X_{max}^7$ | 2.3231 | 0.6845 | 1.9563 | 0.7720 | 0.2601 | 0.2258 | 0.1923 |
|  | $X_{min}^7$ | 0.1921 | 0.1921 | 0.1921 | 0.1921 | 0.1921 | 0.1921 | 0.1916 |
|  | $A_{max}^7$ | 1109.1% | 256.3% | 918.2% | 301.8% | 35.4% | 17.5% | 0.4% |
| N8 | $X_{max}^8$ | 2.8459 | 2.8459 | 2.8459 | 2.8459 | 2.8459 | 2.8459 | 2.8459 |
|  | $X_{min}^8$ | 2.8459 | 2.3042 | 1.6289 | 2.8332 | 2.8282 | 2.6950 | 2.7233 |
|  | $A_{max}^8$ | 0.0% | 23.5% | 74.7% | 0.5% | 0.6% | 5.6% | 4.5% |

Note: A is deck breakage, B is concrete slab stiffness degradation, C is steel beam microcrack, D is diaphragm stiffness degradation, E is stud fracture, F is bearing damage, and G is pier stiffness degradation. N1 is the maximum deflection of the side span, N2 is the maximum stress of the structure, N3 is the maximum displacement of the pier top, N4 is the vertical vibration displacement (VVD) in the middle of the side span, N5 is the vertical vibration acceleration (VVA) in the middle of the side span, N6 is the vertical vibration displacement (VVD) in the middle of midspan, N7 is the vertical vibration acceleration (VVA) in the middle of midspan, and N8 is the first-order modal frequency. $X_{max}^k$ ($k = 1, 2, \ldots, 8$) indicates the maximum value of the $k$-index among the 5 damage grades, $X_{min}^k$ ($k = 1, 2, \ldots, 8$) indicates the minimum value of the $k$-index among the 5 damage grades, and $A_{max}^k$ ($k = 1, 2, \ldots, 8$) denotes the maximum increase in the $k$-index.

### 6.3. Analysis of Sensitivity Influencing Factors

Using the linear cumulative damage hypothesis outlined in Section 5.2 of the paper, we calculated the sensitivity of both static and dynamic indices to the influence of seven types of damage through Equations (37) and (38), and the outcomes are showcased in Table 8. To provide a clearer illustration of how these seven types of damage affect the sensitivity of each static and dynamic index, the results in Table 8 were summarized and are presented in Figure 14.

**Table 8.** Sensitivity of static and dynamic indexes.

| | A | B | C | D | E | F | G |
|---|---|---|---|---|---|---|---|
| N1 | 0.0000 | 0.2686 | 0.6083 | 0.0090 | 0.0093 | 0.0519 | 0.0529 |
| N2 | 0.0000 | 0.1887 | 0.5347 | 0.0724 | 0.0097 | 0.0348 | 0.1596 |
| N3 | 0.0000 | 0.5720 | 0.2326 | 0.0041 | 0.0043 | 0.0225 | 0.1646 |
| N4 | 0.0506 | 0.0502 | 0.7117 | 0.1607 | 0.0069 | 0.0128 | 0.0071 |
| N5 | 0.5166 | 0.0833 | 0.2349 | 0.0936 | 0.0365 | 0.0303 | 0.0047 |
| N6 | 0.0464 | 0.0428 | 0.7344 | 0.1529 | 0.0066 | 0.0119 | 0.0050 |
| N7 | 0.4203 | 0.0971 | 0.3480 | 0.1144 | 0.0134 | 0.0066 | 0.0001 |
| N8 | 0.0000 | 0.2149 | 0.6829 | 0.0041 | 0.0057 | 0.0512 | 0.0412 |

Note: The meanings of A~E and N1~N8 in the table are consistent with Table 6.

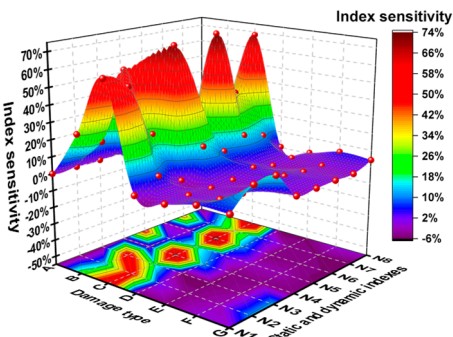

**Figure 14.** Influence of damage on static and dynamic indexes. The meanings of A~E and N1~N8 in the table are consistent with Table 6.

Based on Figure 14, it is clear that the surface map's peak is concentrated on all static and dynamic indicators associated with steel beam microcracks, the maximum displacement of pier tops associated with the deterioration of the concrete slab stiffness, and VVA at the side span and midspan associated with deck breakage. Steel beam microcracks have the greatest impact on the sensitivity of each static and dynamic indicator, with five indicators having sensitivity of 0.5 or higher, including the maximum deflection of the side span, the maximum stress of the structure, VVD at the side span and midspan, and first-order modal frequency, which reach 0.6083, 0.5347, 0.7117, 0.7344, and 0.6829, respectively. Concrete slab stiffness degradation has the greatest impact on the sensitivity of the indicator for the maximum displacement of pier tops, with sensitivity of 0.5720. Deck breakage has the greatest impact on the sensitivity of the indicator for VVA at the side span and midspan, with sensitivities of 0.5166 and 0.4203, respectively.

Based on Table 8, it is evident that deck breakage has the most significant impact on VVA sensitivity in the middle of spans. When the bridge deck is damaged, the sensitivity of VVA in the middle of the side span and midspan is 0.5166 and 0.4203, respectively, while the sensitivity of other static and dynamic indicators remains within 0.06. The degradation of the concrete slab stiffness has a greater impact on the sensitivity of three indicators: maximum deflection of the side span, maximum displacement of pier tops, and first-order modal frequency, with sensitivity values of 0.2686, 0.5720, and 0.2149, respectively. The sensitivity of other indicators under the influence of concrete slab stiffness degradation remains within 0.2. Under the influence of steel beam microcracks, the sensitivity of five indicators, including the maximum deflection of the side span, maximum stress of the structure, VVD of the side span and midspan, and first-order modal frequency, remains above 0.5. The degradation of the diaphragm stiffness has a certain impact on the sensitivity of two indicators, namely the VVD of the side span and midspan, with sensitivity values of 0.1607 and 0.1529, respectively. The sensitivity of other indicators under the influence of diaphragm stiffness degradation remains below 0.12. The sensitivity of various indicators under two types of damage, namely stud fracture and bearing damage, is less than 0.06. Pier stiffness degradation only affects the sensitivity of two indicators to some extent,

namely the maximum stress of the structure and the maximum displacement of pier tops, with sensitivity values of 0.1596 and 0.1646, respectively, while the sensitivity of other indicators remains below 0.06.

Based on the sensitivity calculation results presented in Table 8 for the static–dynamic indicators, and utilizing Equation (39) from Section 5.2 of the paper, we computed the cumulative impact contribution of each damage type on all static–dynamic indicators, $V_i$. The resulting values are illustrated in Figure 15. As depicted in Figure 15, the steel beam microcracks exerted the most substantial impact contribution on the eight static–dynamic indicators, with a total sensitivity impact contribution of 4.0875. On the other hand, the contribution of stud fractures to sensitivity was the smallest, at 0.0924.

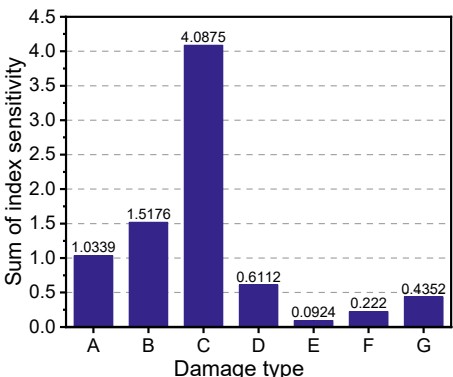

**Figure 15.** The sum of the sensitivity of indexes for different damages. Note: A is deck breakage, B is concrete slab stiffness degradation, C is steel beam microcrack, D is diaphragm stiffness degradation, E is stud fracture, F is bearing damage, and G is pier stiffness degradation.

Based on the results presented in Figure 15, the normalized outcomes were obtained utilizing Equations (40) and (41) in Section 5.2 of the paper. We then calculated the sensitivity influence factors for the seven types of damage separately, as demonstrated in Figure 16. The sensitivity influence factor values for deck breakage, concrete slab stiffness degradation, steel beam microcracks, diaphragm stiffness degradation, stud fracture, bearing damage, and pier stiffness degradation were found to be 0.13, 0.19, 0.51, 0.08, 0.01, 0.03, and 0.05, respectively. According to the proposed damage sensitivity analysis method in this paper, the magnitude order of sensitivity for the seven types of damage affecting the overall mechanical performance of the steel–concrete composite continuous beam bridge is as follows: steel beam microcracks, concrete slab stiffness degradation, deck breakage, diaphragm stiffness degradation, pier stiffness degradation, bearing damage, and stud fracture.

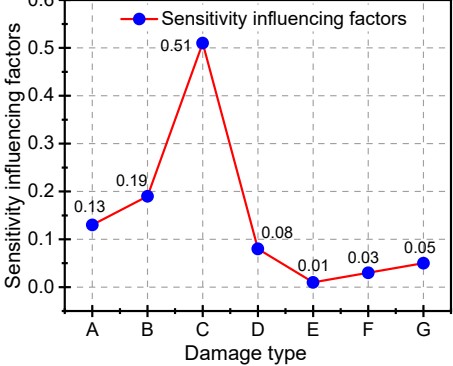

**Figure 16.** Influence factors for each damage sensitivity. Note: A is deck breakage, B is concrete slab stiffness degradation, C is steel beam microcrack, D is diaphragm stiffness degradation, E is stud fracture, F is bearing damage, and G is pier stiffness degradation.

The damage sensitivity analysis of the SCCBB proposed in this paper has high efficiency, and it only takes a few minutes to calculate the sensitivity impact factor when the static and dynamic analysis data are known. However, the specific time required depends on the number of mechanical indexes selected and the number of damage types considered. When there are more mechanical indexes and damage types selected, the calculation time will increase accordingly. The final computational efficiency also depends on the configuration of the computer and the computational efficiency of the numerical simulation software needed to calculate the mechanical index.

## 7. Conclusions

This paper presents a damage sensitivity analysis method for steel–concrete composite continuous beam bridges using the ET algorithm. The study is conducted on a three-span bridge and investigates the sensitivity of three static indicators and five dynamic indicators to seven types of damage. These damage types include deck breakage, concrete slab stiffness degradation, steel beam microcracks, diaphragm stiffness degradation, stud fracture, bearing damage, and pier stiffness degradation. The finite element method is employed to analyze the mechanical performance of the bridge under each type of damage. The findings of this study are presented as follows.

1. The local stiffness reduction method and modification of element real constants were used to establish five damage grades for each of the seven damage types. Based on the analysis of three static and five dynamic indicators, sensitivity factors were calculated for deck breakage, concrete slab stiffness degradation, steel beam microcracks, diaphragm stiffness degradation, stud fracture, bearing damage, and pier stiffness degradation. The sensitivity factors for each damage type were found to be 0.13, 0.19, 0.51, 0.08, 0.01, 0.03, and 0.05, respectively. It can be observed that the sensitivity of the seven types of damage to the mechanical performance of the steel–concrete composite continuous beam bridge decreases in the following order: steel beam microcracks, concrete slab stiffness degradation, deck breakage, diaphragm stiffness degradation, pier stiffness degradation, bearing damage, and stud fracture. In the future, the safety, durability, and damage state of steel beams and concrete slab structures should be paid sufficient attention in the design, construction, and testing of SCCBBs.

2. The damage sensitivity analysis method for the SCCBB based on the ET algorithm establishes a relationship between various types and grades of damage and multiple static and dynamic indicators. Through scientific mathematical deduction, the sensitivity impact factors of different types of damage on the mechanical performance of SCCBBs are obtained, which allows for the quantification of SCCBBs' damage sensitivity. This method guides the damage classification of SCCBBs in a state of multiple interlaced damages. It can provide a theoretical basis for the formulation of a flexible traffic operation strategy for SCCBBs after disasters.

3. Considering the numerical computation cost and the convergence of the results, this paper selected only seven types of damage, five damage grades, three static indicators, and five dynamic indicators to simulate the degradation of structural stiffness using a reduced material elastic modulus. The simplified damage simulation method and limited mechanical indicators imply that the calculation results have certain limitations. In future research, a broader range of damage types, damage grades, and mechanical indicators could be considered. Sophisticated finite element simulation methods could be employed to simulate different structural damages, and more types of steel–concrete composite structure bridges could be analyzed to enhance the scientific and accurate calculation results of damage sensitivity in SCCBBs.

4. The damage sensitivity analysis method for SCCBBs based on the ET algorithm assumes a linear relationship among multiple damage types and does not account for energy dissipation caused by damping. However, in reality, multiple damage types are interdependent and coupled. Therefore, future research should focus on developing nonlinear damage sensitivity analysis methods that consider the influence

of damping, which will enhance the engineering applicability of the steel–concrete composite structure bridge damage sensitivity analysis method.

**Author Contributions:** Conceptualization, Z.G. and J.Z.; methodology, Z.G., J.B. and J.Z.; software, Z.G.; validation, Z.G. and J.Z.; formal analysis, Z.G. and J.Z.; investigation, Z.G.; resources, Z.G. and J.B.; data curation, Z.G.; writing—original draft preparation, Z.G., J.Z. and W.C.; writing—review and editing, Z.G., J.B., J.Z. and W.C.; visualization, Z.G. and J.Z.; supervision, J.B.; project administration, J.B. and X.H.; funding acquisition, J.B. All authors have read and agreed to the published version of the manuscript.

**Funding:** This research was funded by the National Key R&D Program of China (grant No. 2021YFB2600600, 2021YFB2600605), the Key R&D Program of Hebei Province (grant No. 19275405D), the Science and Technology Project of Hebei Education Department (grant No. QN2023067), and the Postgraduate Innovation Fund Project of Hebei Province (grant No. CXZZSS2023078).

**Data Availability Statement:** All data analyzed in this study have been included in this paper.

**Conflicts of Interest:** The authors declare no conflict of interest.

## Appendix A

**Table A1.** The names and details of the symbols used in the paper.

| Symbols | Meaning |
| --- | --- |
| $A_e$ | The bearing area. |
| $A_s$ | The cross-sectional area of the stud connector rod. |
| $A_{i,max}^{k}$ | The maximum increase value of the $k$th index of type $i$ damage. |
| $C$ | The damping coefficient vector of the vehicle. |
| $C_b$ | The damping matrix of the bridge. |
| $C_v$ | The damping matrices of the vehicle. |
| $C_{uL}^{i}$ | The suspension damping coefficients of the left sides of the vehicle. |
| $C_{uR}^{i}$ | The suspension damping coefficients of the right sides of the vehicle. |
| $C_{lL}^{i}$ | The tire damping coefficients of the left sides of the vehicle. |
| $C_{lR}^{i}$ | The tire damping coefficients of the right sides of the vehicle. |
| $d_b$ | The displacement matrix of the bridge. |
| $d_{ss}$ | The diameter of the stud connector. |
| $d_0$ | The diameter of the circular reinforcing steel plate in millimeters. |
| $d_v$ | The freedom vector of the vehicle. |
| $EI(x)$ | The stiffness before reduction. |
| $E$ | The compressive elastic modulus. |
| $E_c$ | The elastic modulus of concrete. |
| $E_e$ | The compressive modulus of elasticity. |
| $E_s$ | The elastic modulus of the stud connector material. |
| $E_i$ | The overall elastic modulus of the corresponding structure when it is damaged at grade $i$. |
| $f_{ck}$ | The standard compressive strength of concrete. |
| $F_{bv}$ | The moment matrix of the forces applied by a three-axis vehicle on the bridge deck. |
| $F_{vb}$ | The load vector of the vehicle–bridge interaction force caused by wheel deformation. |
| $f_{vg}$ | The load vector caused by the self-weight of the vehicle. |
| $G_{ij}$ | The weighted impurity. |
| $G(n)$ | The displacement power spectral density. |
| $G_d(n_0)$ | The bridge grading factor. |
| $G_e$ | The shear modulus in megapascals. |
| $H(X)$ | The impurity function. |
| $J_{yz}^{i}$ | The moment of inertia of vehicle body roll characteristics. |
| $J_{zx}^{i}$ | The moment of inertia of the pitching characteristics of the vehicle body. |
| $k_1$ | The shear correction factors whose values depend on the cross-sectional shape of the substructure. |
| $k_2$ | The shear correction factors whose values depend on the cross-sectional shape of the substructure. |
| $K_{uL}^{i}$ | The suspension stiffness coefficients of the left side of the vehicle. |

**Table A1.** *Cont.*

| Symbols | Meaning |
| --- | --- |
| $K_{uR}^i$ | The suspension stiffness coefficients of the right side of the vehicle. |
| $K_{lL}^i$ | The tire stiffness coefficients of the left sides of the vehicle. |
| $K_{lR}^i$ | The tire stiffness coefficients of the right sides of the vehicle. |
| $k_{ss}$ | The shear stiffness of the stud connector. |
| $k_v$ | The axial stiffness of the stud connector. |
| $K_i$ | The spring stiffness constant corresponding to class $i$ damage. |
| $K$ | The vehicle stiffness coefficient vector. |
| $K_b$ | The stiffness matrix of the bridge. |
| $K_v$ | The stiffness matrices of the vehicle. |
| $l_c$ | The distance from the center of the crack zone to the left node of the element. |
| $l_{0a}$ | The length of the rectangular reinforcing steel plate in millimeters. |
| $l_{0b}$ | The width of the rectangular reinforcing steel plate in millimeters. |
| $L'$ | The element length. |
| $l$ | The length of the stud connector rod. |
| $M$ | The mass (moment of inertia) vector of the vehicle. |
| $M_v$ | The mass matrices of the vehicle. |
| $M_b$ | The mass matrix of the bridge. |
| $M_{aL}^i$ | The tire masses of the left sides of the vehicle. |
| $M_{aR}^i$ | The tire masses of the right sides of the vehicle. |
| $n_0$ | The reference spatial frequency. |
| $n$ | The spatial frequency ($m^{-1}$). |
| $n_u$ | The upper limits of the effective spatial frequency. |
| $n_d$ | The lower limits of the effective spatial frequency. |
| $N$ | The total number of sampling points. |
| $S_1$ | The shape factors of rectangular and circular rubber bearings. |
| $S_2$ | The shape factors of rectangular and circular rubber bearings. |
| $t + \Delta t$ | The next step in the integration process. |
| $t$ | The current step in the integration process. |
| $t_1$ | The thickness of the middle rubber layer in millimeters. |
| $V_i^k$ | The $k$th index sensitivity of type $i$ damage. |
| $V_i$ | The contribution of the influence of type $i$ damage on all mechanical indexes. |
| $\mu_i$ | Sensitivity impact factor. |
| $V_{IMx}$ | The sensitivity coefficient of the importance measures $IMx$ in the ET forest. |
| $V_{IMx,k,m}$ | The sensitivity coefficient of strength indicator $IMx$ at node $m$ in set $M$. |
| $w$ | Frequency. |
| $W^e$ | The work exerted by external forces. |
| $X_{i,max}^k$ | The maximum values of the $k$th index for type $i$ damage at different damage grades. |
| $x$ | The longitudinal coordinate value of the bridge deck. |
| $X$ | The displacement vector containing the overall node. |
| $X_{ij}$ | All data sets before the split at node $v_{ij}$. |
| $X_{i,min}^k$ | The minimum values of the $k$th index for type $i$ damage at different damage grades. |
| $Y$ | The displacement at the contact position between the wheel and the bridge. |
| $Z_{aL}^i$ | The vertical displacements of the tires on the left sides of the vehicle. |
| $Z_{aR}^i$ | The vertical displacements of the tires on the right sides of the vehicle. |
| $Z_{r1}$ | The vertical displacements of the front end. |
| $Z_{r2}$ | The vertical displacements of the body. |
| $\alpha$ | Damage parameters. |
| $\beta$ | Damage parameters. |
| $\gamma$ | The weight of the linear change between the initial and final acceleration effects on the velocity change. |
| $\xi_c$ | The damping ratio. |
| $\theta_{r1}$ | The pitch angle displacements of the front end. |
| $\theta_{r2}$ | The pitch angle displacements of the body. |
| $\Phi_{r1}$ | The lateral tilt angle displacements of the front end. |
| $\Phi_{r2}$ | The lateral tilt angle displacements of the body. |
| $\varphi_k$ | The random phase angle. |

**Table A2.** Acronyms.

| Noun | Acronym |
| --- | --- |
| steel–concrete composite beam bridge | SCCBB |
| extremely randomized trees | ET |
| Engineered Cementitious Composite | ECC |
| finite element method | FEM |
| two-dimensional | 2D |
| three-dimensional | 3D |
| vertical vibration displacement | VVD |
| vertical vibration acceleration | VVA |

**Appendix B**

The submatrices in Equations (8) and (9) are shown as follows.

$$
K_{11} = \begin{bmatrix} K_{lL}^1 - K_{uL}^1 & & & \\ & K_{lR}^1 - K_{uR}^1 & & \\ & & \ddots & \\ & & & K_{lR}^3 - K_{uR}^3 \end{bmatrix} \cdot C_{11} = \begin{bmatrix} C_{lL}^1 - C_{uL}^1 & & & \\ & C_{lR}^1 - C_{uR}^1 & & \\ & & \ddots & \\ & & & C_{lR}^3 - C_{uR}^3 \end{bmatrix}
$$

$$
K_{12} = \begin{bmatrix} -K_{uL}^1 & -bK_{uL}^1 & K_{uL}^1 L_1 & 0 & 0 \\ -K_{uR}^1 & bK_{uR}^1 & -L_2 K_{uL}^2 & 0 & 0 \\ -K_{uL}^2 & -bK_{uL}^2 & -L_2 K_{uL}^2 & 0 & 0 \\ -K_{uR}^2 & -bK_{uR}^2 & -L_2 K_{uR}^2 & 0 & 0 \\ -\frac{L_4}{L_6} K_{uL}^3 & 0 & \frac{L_4 L_5}{L_6} K_{uL}^3 & -\frac{L_4 + L_6}{L_6} K_{uL}^3 & -bK_{uL}^3 \\ -\frac{L_4}{L_6} K_{uR}^3 & 0 & \frac{L_4 L_5}{L_6} K_{uR}^3 & -\frac{L_4 + L_6}{L_6} K_{uR}^3 & bK_{uR}^3 \end{bmatrix} \cdot C_{12} = \begin{bmatrix} -C_{uL}^1 & -bC_{uL}^1 & C_{uL}^1 L_1 & 0 & 0 \\ -C_{uR}^1 & bC_{uR}^1 & -L_2 C_{uL}^2 & 0 & 0 \\ -C_{uL}^2 & -bC_{uL}^2 & -L_2 C_{uL}^2 & 0 & 0 \\ -C_{uR}^2 & -bC_{uR}^2 & -L_2 C_{uR}^2 & 0 & 0 \\ -\frac{L_4}{L_6} C_{uL}^3 & 0 & \frac{L_4 L_5}{L_6} C_{uL}^3 & -\frac{L_4 + L_6}{L_6} C_{uL}^3 & -bC_{uL}^3 \\ -\frac{L_4}{L_6} C_{uR}^3 & 0 & \frac{L_4 L_5}{L_6} C_{uR}^3 & -\frac{L_4 + L_6}{L_6} C_{uR}^3 & bC_{uR}^3 \end{bmatrix}
$$

$$
K_{21} = \begin{bmatrix}
-K^1_{uL} & -K^1_{uR} & -K^2_{uL} & -K^2_{uR} & -K^3_{uL} & -K^3_{uR} \\
-bK^1_{uL} & bK^1_{uR} & -bK^2_{uL} & bK^2_{uR} & 0 & 0 \\
(L_1+L_2)K^1_{uL} & (L_1+L_5)K^1_{uR} & (L_5-L_2)K^2_{uL} & (L_5-L_2)K^2_{uR} & 0 & 0 \\
L_6 K^1_{uL} & L_6 K^1_{uR} & L_6 K^2_{uL} & L_6 K^2_{uR} & -L_4 K^3_{uL} & -L_4 K^3_{uR} \\
0 & 0 & 0 & 0 & -bK^3_{uL} & bK^3_{uR}
\end{bmatrix}
\cdot
C_{21} = \begin{bmatrix}
-C^1_{uL} & -C^1_{uR} & -C^2_{uL} & -C^2_{uR} & -C^3_{uL} & -C^3_{uR} \\
-bC^1_{uL} & bC^1_{uR} & -bC^2_{uL} & bC^2_{uR} & 0 & 0 \\
(L_1+L_2)C^1_{uL} & (L_1+L_5)C^1_{uR} & (L_5-L_2)C^2_{uL} & (L_5-L_2)C^2_{uR} & 0 & 0 \\
L_6 C^1_{uL} & L_6 C^1_{uR} & L_6 C^2_{uL} & L_6 C^2_{uR} & -L_4 C^3_{uL} & -L_4 C^3_{uR} \\
0 & 0 & 0 & 0 & -bC^3_{uL} & bC^3_{uR}
\end{bmatrix}
$$

$$
K_{22} = \begin{bmatrix}
K^1_{uL}+K^1_{uR}+K^2_{uL}+K^2_{uR}-\frac{L_4}{L_5}\left(K^3_{uL}+K^3_{uR}\right) & b\left(K^1_{uL}-K^1_{uR}+K^2_{uL}-K^2_{uR}\right) & L_1\left(K^1_{uL}+K^1_{uR}\right)-L_2\left(K^2_{uL}+K^2_{uR}\right)-\frac{L_4 L_5}{L_6}\left(K^3_{uL}+K^3_{uR}\right) & \frac{(L_4+L_6)}{L_6}\left(K^3_{uL}+K^3_{uR}\right) & b\left(K^3_{uL}-K^3_{uR}\right) \\
b\left(K^1_{uL}-K^1_{uR}+K^2_{uL}-K^2_{uR}\right) & b^2\left(K^1_{uL}+K^1_{uR}+K^2_{uL}+K^2_{uR}\right) & b\left[L_1\left(-K^1_{uL}+K^1_{uR}\right)+L_2\left(K^2_{uL}-K^2_{uR}\right)\right] & 0 & 0 \\
-(L_1+L_5)\left(K^1_{uL}+K^1_{uR}\right)+(L_2-L_5)\left(K^2_{uL}+K^2_{uR}\right) & b\left[(L_1+L_5)\left(-K^1_{uL}+K^1_{uR}\right)+(L_2-L_5)\right]\left(K^2_{uL}-K^2_{uR}\right) & L_2(L_2-L_5)\left(K^2_{uL}+K^2_{uR}\right)+L_1(L_1+L_5)\left(K^1_{uL}+K^1_{uR}\right) & 0 & 0 \\
-\frac{L_4^2}{L_5}\left(K^3_{uL}+K^3_{uR}\right)-L_6\left(K^1_{uL}+K^1_{uR}+K^2_{uL}+K^2_{uR}\right) & bL_6\left(-K^1_{uL}+K^1_{uR}-K^2_{uL}+K^2_{uR}\right) & L_1 L_6\left(K^1_{uL}+K^1_{uR}\right)-L_2 L_6\left(K^2_{uL}+K^2_{uR}\right)-\frac{L_4^2 L_5}{L_6}\left(K^3_{uL}+K^3_{uR}\right) & \frac{L_4(L_4+L_6)}{L_6}\left(K^3_{uL}+K^3_{uR}\right) & bL_4\left(K^3_{uL}-K^3_{uR}\right) \\
\frac{bL_4}{L_6}\left(-K^3_{uL}+K^3_{uR}\right) & 0 & \frac{bL_4 L_5}{L_6}\left(K^3_{uR}-K^3_{uL}\right) & \frac{b(L_4+L_6)}{L_6}\left(K^3_{uL}-K^3_{uR}\right) & b^2\left(K^3_{uL}+K^3_{uR}\right)
\end{bmatrix}
$$

$$
C_{22} = \begin{bmatrix}
C^1_{uL}+C^1_{uR}+C^2_{uL}+C^2_{uR}-\frac{L_4}{L_5}\left(C^3_{uL}+C^3_{uR}\right) & b\left(C^1_{uL}-C^1_{uR}+C^2_{uL}-C^2_{uR}\right) & L_1\left(C^1_{uL}+C^1_{uR}\right)-L_2\left(C^2_{uL}+C^2_{uR}\right)-\frac{L_4 L_5}{L_6}\left(C^3_{uL}+C^3_{uR}\right) & \frac{(L_4+L_6)}{L_6}\left(C^3_{uL}+C^3_{uR}\right) & b\left(C^3_{uL}-C^3_{uR}\right) \\
b\left(C^1_{uL}-C^1_{uR}+C^2_{uL}-C^2_{uR}\right) & b^2\left(C^1_{uL}+C^1_{uR}+C^2_{uL}+C^2_{uR}\right) & b\left[L_1\left(-C^1_{uL}+C^1_{uR}\right)+L_2\left(C^2_{uL}-C^2_{uR}\right)\right] & 0 & 0 \\
-(L_1+L_5)\left(C^1_{uL}+C^1_{uR}\right)+(L_2-L_5)\left(C^2_{uL}+C^2_{uR}\right) & b\left[(L_1+L_5)\left(-C^1_{uL}+C^1_{uR}\right)+(L_2-L_5)\right]\left(C^2_{uL}-C^2_{uR}\right) & L_2(L_2-L_5)\left(C^2_{uL}+C^2_{uR}\right)+L_1(L_1+L_5)\left(C^1_{uL}+C^1_{uR}\right) & 0 & 0 \\
-\frac{L_4^2}{L_5}\left(C^3_{uL}+C^3_{uR}\right)-L_6\left(C^1_{uL}+C^1_{uR}+C^2_{uL}+C^2_{uR}\right) & bL_6\left(-C^1_{uL}+C^1_{uR}-C^2_{uL}+C^2_{uR}\right) & L_1 L_6\left(C^1_{uL}+C^1_{uR}\right)-L_2 L_6\left(C^2_{uL}+K^2_{uR}\right)-\frac{L_4^2 L_5}{L_6}\left(C^3_{uL}+C^3_{uR}\right) & \frac{L_4(L_4+L_6)}{L_6}\left(C^3_{uL}+C^3_{uR}\right) & bL_4\left(C^3_{uL}-C^3_{uR}\right) \\
\frac{bL_4}{L_6}\left(-C^3_{uL}+C^3_{uR}\right) & 0 & \frac{bL_4 L_5}{L_6}\left(C^3_{uR}-C^3_{uL}\right) & \frac{b(L_4+L_6)}{L_6}\left(C^3_{uL}-C^3_{uR}\right) & b^2\left(C^3_{uL}+C^3_{uR}\right)
\end{bmatrix}
$$

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
