# Peer review of "Theoretical and Numerical Investigation of Damage Sensitivity of Steel–Concrete Composite Beam Bridges"

_buildings, doi:10.3390/buildings13051109_

Round 1
Reviewer 1 Report
The study considers the damage sensitivity of steel-concrete composite beam bridges using the extremely randomized trees (ET) algorithm. The description in the manuscript is not rigorous. The re-production of the results is not impossible. The reviewer does not recommend the publication of this paper.
1. The derivation of governing equation in Section 2 is not complete. The reader must read other papers to understand the formulation. A complete formulation must be provided in the paper.
2. The numerical model in Sections 2.2 and 2.3 is different from the formulation in Section 2.1. Which one is considered in this study?
3. The authors propose to consider the effects of deck breakage in terms of the surface irregularity in Section 3.1. But, the quantitative description cannot be found. For example, how can the effects of the diameter and depth of potholes on a deck be considered in the PSD function in Eq. (8)?
4. In Section 3.2, three models for concrete structure damage are provided. Which model is considered in this study?
5. How is the steel structure damage represented? It is not described in Section 3.1.
6. The hysteretic behavior is very important for the rubber bearing. Thus, it must be represented by hysteretic elements, not by linear spring elements.
7. Is the reduction of Young's modulus in Eq. (21) applied to the whole structure or to damaged parts?
8. Eqs. (21) and (22) just show numerical reductions of the sitffness. How can the reductions are related to the real damage of the considered structure?
9. Section 4.1 is not sufficient to describe the ET method. Reference 25 cannot be found. The reviewer recommends to include more references or detailed formulation for the method.
10. Line 414: the impact of each type of damage on the static and dynamic indicators of the bridge is assumed to be linearly related. Is this assumption correct? The damage usually influences static and dynamic properties of structures in nonlinear manner. The assumption must be proven.
11. Eqs. (29) to (31) are not correct. The numbers of damages and indices must be considered in the equations.
Reviewer 2 Report
The article is well documented, and numerical analyses are detailed presented. Nevertheless, the publication in the “Buildings, MDPI” is not recommended unless the following suggestions are taken into account:
A) The importance and research significance of the article needs to be better highlighted within abstract, introduction and conclusions.
B) Introduction: In the few decades, steel-concrete composite beam bridges have widely been used to avoid the long-term phenomena of concrete, such as tendon relaxation, creep, curing, and shrinkage, which are typical of concrete bridge-girders. So, it is advised to refer about this important issue through the following references:
- https://doi.org/10.1061/(ASCE)BE.1943-5592.0001489
- https://doi.org/10.1016/j.istruc.2021.10.093
C) Please, insert a table/s which list/s the types of finite element used, with the corresponding amounts within the finite element model and mesh sizes.
D) Section 3.2: Concrete cracks have been modeled by deleting specific amounts of finite element meshes. Has this way already been used before ? If yes, please cite the corresponding work/s existing in the literature.
E) Percentage errors (across the whole paper) should be reported using one decimal.
F) Please substitute the term “mid-span” with “midspan” within the text.
G) Please cite the finite element software ANSYS in the references.
H) An “Appendix” section, containing names and elaboration of the symbols used, should be inserted at the beginning or at the end of the article.
Reviewer 3 Report
This research paper presents a novel approach to detect structural damages based on sensitivity analysis. The methodology is implemented in steel-concrete composite continuous beam bridges using the ET algorithm. In general, the paper is well-structured and seems to be contributing to the Buildings Journal. However, the following major revisions must be addressed by the Authors before proceeding further.
1. By the end of the Abstract Section, please document the main findings of this paper.
2. In the Introduction Section, the discussion about the damage detection techniques in bridges must be expanded. For example, a discussion must be incorporated in terms of the use of different devices and techniques to detect structural damage. In this sense, please consider the following technical papers and some others in the discussion.
a. https://doi.org/10.1016/j.engfailanal.2015.07.030
b. https://doi.org/10.1002/stc.1852
c. https://doi.org/10.3390/s18010262
d. https://doi.org/10.1002/stc.2264
e. https://doi.org/10.1155/2019/6429430
a. Etc…
3. In the last part of the Introduction Section, please include the main contribution of this paper to the Buildings Journal.
4. Figure 3 presents the Finite Element Model of the bridge under consideration. Within this frame of reference, what finite element software was used? In addition, please justify the implementation of such a finite element software in this paper. It seems that the finite element software implemented in this manuscript is ANSYS but not quite sure about this. Please clarify this point. Thanks.
5. By the end of the first paragraph of Section 3, the Authors are declaring the use of 35 damage scenarios for numerical analysis models. Please justify in a more clear way the use of such 35 damage scenarios.
6. Please document more information about Table 1. In other words, explain more into detail the use of Table 1 in this paper.
7. It seems that Equation (12) is involving three equations, please separate those equations one by one in the manuscript.
8. In Section 3.3, please include a figure that illustrates the steel structure damage technique to be used.
9. In some part of the paper, the Authors have to include a list of acronyms. There are a lot of them in the manuscript.
10. Before the discussion of the results of the research developed in this paper. The Authors must include a general flowchart of the process of the methodology implemented in the paper.
11. It seems that the method presented by the Authors in this paper is quite accurate, however, how efficient is it? In other words, how much computational time are taking the analysis presented in the paper?
12. Based on the methodology and results presented by the Authors of this paper, how far is this approach to be included in major guidelines about the retrofit of bridge structures? in particular to detect damages.
13. In the Conclusions, please include the main limitation(s) of the research presented in this paper.
14. In the References Section, please include the DOI of every citation.
Round 2
Reviewer 1 Report
1. It is mentioned in the answer that the purpose of Section 2.1 is to let readers have a clearer understanding of the vehicle-bridge coupling vibration theory, which mainly focuses on steel-concrete composite beam bridges considering slip. However, the formulation is not implemented in this study. The beam elements in Section 2.1 is not considered in this study. The numerical model is constructed with the finite elements in ANSYS. Why is Section 2.1 necessary? If the authors want to present a basic theory for the vehicle-bridge interaction, it must be described in a consistent way with the finite-element model.
2. The reviewer cannot find any clear answer for the deck breakage. The simulations of random process based on its PSD function are very widely used in a variety of areas. The approach is very common. However, the manuscript does not present any quantitative description for how to consider the deck breakage in the PSD function. The description must be included.
3. The numerical model for hysteretic rubber bearings are widely considered in other studies. For example, simple bi-linear spring can be considered. A nonlinear model must be considered for the hysteretic rubbel bearings.
4. The reduced Young's modulus must be used just for local damaged areas. This is very similar to the smeared crack model. It does not make sense to reduce Young's modulus of the whole structure because most parts of the structure is not damaged.
5. This journal is an internatioanl journal. The references must be accessible to every readers. English references are preferred to chiniese ones. Some of the references must be replaced.
Reviewer 2 Report
The authors have adequately addressed my comments.
Author Response
Dear Reviewer,
On behalf of all the authors, I would like to express our sincere gratitude for your valuable feedback during the review process. We highly appreciate your insightful comments and have carefully revised and improved our paper based on your suggestions.Through rigorous revisions and enhancements, the quality of our paper has been significantly improved, and the logic has become more coherent. We once again thank you for your valuable comments on our paper, and we would be more than happy to answer any further questions or assist you in any way possible.
Best regards.
Reviewer 3 Report
The Authors have addressed all the recommendations. The paper is now in a more suitable form to be considered for publication. Thanks for addressing the revisions! Good job!
Author Response
Dear Reviewer,
On behalf of all the authors, I would like to express our sincere gratitude for your valuable feedback during the review process. We highly appreciate your insightful comments and have carefully revised and improved our paper based on your suggestions. Through rigorous revisions and enhancements, the quality of our paper has been significantly improved, and the logic has become more coherent. We once again thank you for your valuable comments on our paper, and we would be more than happy to answer any further questions or assist you in any way possible.
Best regards.
Round 3
Reviewer 1 Report
The reviewer believes that the manuscript is revised properly and can be published in the journal.